

# DNA barcode data accurately assign higher spider taxa

Jonathan A. Coddington[1], Ingi Agnarsson[1,2], Ren-Chung Cheng[3], Klemen Čandek[3], Amy Driskell[1], Holger Frick[4], Matjaž Gregorič[3], Rok Kostanjšek[5], Christian Kropf[4], Matthew Kweskin[1], Tjaša Lokovšek[3], Miha Pipan[3,6], Nina Vidergar[3] and Matjaž Kuntner[1,3]

[1] National Museum of Natural History, Smithsonian Institution, Washington, D.C., United States
[2] Department of Biology, University of Vermont, Burlington, Vermont, United States
[3] EZ Lab, Institute of Biology, Research Centre of the Slovenian Academy of Sciences and Arts, Ljubljana, Slovenia
[4] Department of Invertebrates, Natural History Museum Bern, Bern, Switzerland
[5] Department of Biology, Biotechnical Faculty, University of Ljubljana, Ljubljana, Slovenia
[6] Department of Biochemistry, University of Cambridge, Cambridge, United Kingdom

Corresponding author
Matjaž Kuntner, kuntner@gmail.com

## ABSTRACT

The use of unique DNA sequences as a method for taxonomic identification is no longer fundamentally controversial, even though debate continues on the best markers, methods, and technology to use. Although both existing databanks such as GenBank and BOLD, as well as reference taxonomies, are imperfect, in best case scenarios "barcodes" (whether single or multiple, organelle or nuclear, loci) clearly are an increasingly fast and inexpensive method of identification, especially as compared to manual identification of unknowns by increasingly rare expert taxonomists. Because most species on Earth are undescribed, a complete reference database at the species level is impractical in the near term. The question therefore arises whether unidentified species can, using DNA barcodes, be accurately assigned to more inclusive groups such as genera and families—taxonomic ranks of putatively monophyletic groups for which the global inventory is more complete and stable. We used a carefully chosen test library of CO1 sequences from 49 families, 313 genera, and 816 species of spiders to assess the accuracy of genus and family-level assignment. We used BLAST queries of each sequence against the entire library and got the top ten hits. The percent sequence identity was reported from these hits (PIdent, range 75–100%). Accurate assignment of higher taxa (PIdent above which errors totaled less than 5%) occurred for genera at PIdent values >95 and families at PIdent values ≥ 91, suggesting these as heuristic thresholds for accurate generic and familial identifications in spiders. Accuracy of identification increases with numbers of species/genus and genera/family in the library; above five genera per family and fifteen species per genus all higher taxon assignments were correct. We propose that using percent sequence identity between conventional barcode sequences may be a feasible and reasonably accurate method to identify animals to family/genus. However, the quality of the underlying database impacts accuracy of results; many outliers in our dataset could be attributed to taxonomic and/or sequencing errors in BOLD and GenBank. It seems that an accurate and complete reference library of families and genera of life *could* provide accurate higher level taxonomic identifications cheaply and accessibly, within years rather than decades.

## INTRODUCTION

Accurate identification of biological specimens has always limited the application of biological data to important societal problems. Obstacles are well-known and difficult: the vast majority of species are undescribed scientifically (*Erwin, 1982*; *May, 1992*; *Mora et al., 2011*); some unknown but large fraction of higher taxa are not monophyletic (*Goloboff et al., 2009*; *Pyron & Wiens, 2011*); many species can only be identified if certain life stages are available, e.g., adults (*Coddington & Levi, 1991*), classical data sources such as morphology imperfectly track species identity; the discipline of taxonomy continues to dwindle (*Agnarsson & Kuntner, 2007*); the classical process of taxonomic identification is mostly manual and cannot scale to provide the amounts of data required for real-time decisions such as environmental monitoring, invasive species, climate change, etc.

DNA sequence data potentially can eliminate most of these obstacles. DNA barcoding uses a fragment of the mitochondrial gene cytochrome *c* oxidase subunit I (CO1) as a unique species diagnosis/identification tool in the animal kingdom (*Hebert et al., 2003*), with analogous single to several locus protocols applied for vascular plants, ferns, mosses, algae and fungi (*Saunders, 2005*; *Kress & Erickson, 2007*; *Nitta, 2008*; *Chase & Fay, 2009*; *Liu et al., 2010*), protists (*Scicluna, Tawari & Clark, 2006*), and prokaryotes (*Barraclough et al., 2009*). Due to relative ease and inexpensive sequencing, DNA barcoding is a popular tool in species identification and taxonomic applications (e.g., *Doña et al., 2015*; *Xu et al., 2015*; see also *Collins & Cruickshank, 2013*), and the method is no longer fundamentally controversial at the species level (*Pentinsaari, Hebert & Mutanen, 2014*; *Lopardo & Uhl, 2014*; *Čandek & Kuntner, 2015*; *Anslan & Tedersoo, 2015*; *Wang et al., 2015*).

While most species remain undescribed, the situation is not so dire for larger monophyletic groups such as clades accorded the Linnaean ranks of genus or family. In assessing the state of knowledge about biodiversity, it is important to distinguish between the first scientific discovery of an exemplar of a lineage, and phylogenetic understanding of that lineage. Phylogenetic understanding—both tree topology and consequent taxonomic changes, are research programs with no clear end in sight. Linnaean rank is partially arbitrary, and one expects that the number of higher taxa will probably increase over time as understanding improves. Discovery, however, can have an objective definition: the year of the earliest formal taxonomic description of a member of the lineage or taxonomic group in which it is currently included. By this definition the earliest possible discovery of an animal lineage is 1758 (*Linnaeus, 1758*), or in the case of spiders, 1757 (*Clerck, 1757*).

More illuminating are the latest discoveries of lineages with the rank of family within larger clades, because the data tell us something about progress towards broad scale knowledge of biodiversity. The species representing the most recent discovery of a family of birds, for example, is the Broad-billed Sapayoa, *Sapayoa aenigma* Hunt, 1903 (Sapayoaidae). The species representing the most recently discovered mammal family is Kitti's hog-nosed bat, *Craseonycteris thonglongyai* Hill, 1974 (Craseonycteridae). For flowering plants, it is
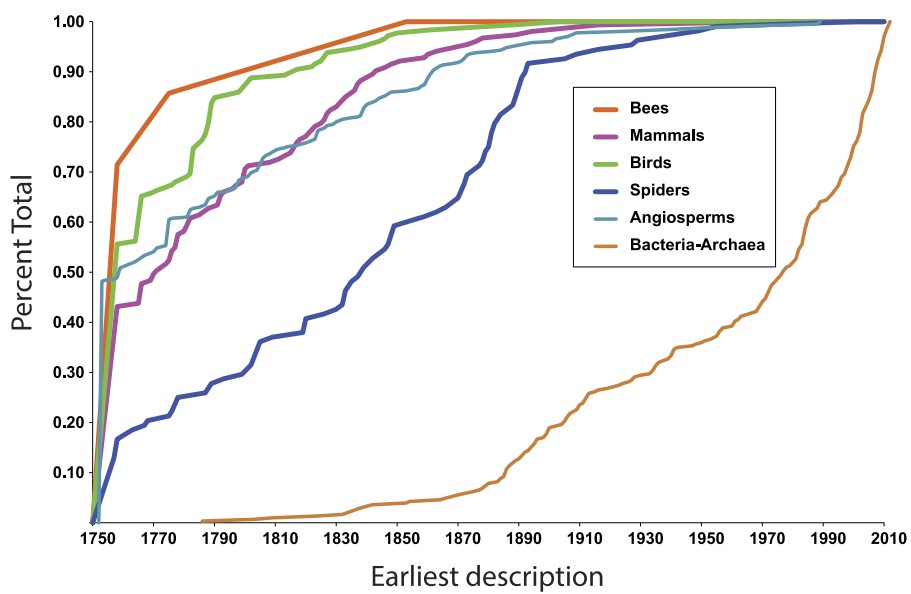

**Figure 1 First discovery of major clades of life.** Accumulation curve of dates of first discovery (year of first description of a contained species) of families for six major clades of life, 1758–2010.

*Gomortega keule* (Molina) Baill, 1972 (Gomertegaceae). For bees, it is *Stenotritus elegans* Smith, 1853 (Stenotritidae). For spiders, a megadiverse and poorly known group, it is *Trogloraptor marchingtoni* Griswold, Audisio & Ledford, 2012 (Trogloraptoridae), but the second most recent discovery of an unambiguously new spider family was in 1955, Gradungulidae (*Forster, 1955*). Figure 1 illustrates the tempo of first discovery of families for these five well-known clades. At the family level, these curves are essentially asymptotic, implying that science is close to completing the inventory of clades ranked as families for these large lineages. On the other hand, for Bacteria and Archaea (Fig. 1), as one would expect, the curve is not asymptotic at all but sharply increasing; prokaryote discovery and understanding is obviously just beginning.

In fact, although many new eukaryote families are named every year, the vast majority of these new names result from advances in phylogenetic understanding, not biological discovery of major new forms of life. The last ten years of Zoological Record suggests that roughly 5–10 truly new families are discovered per year.

In the context of the above question—approximate taxonomic assignment of organisms using DNA sequences—these data suggest that our knowledge of major clades of life is approaching completion. The Global Genome Initiative (GGI; http://ggi.si.edu/) of the Smithsonian Institution via the GGI Knowledge Portal (http://ggi.eol.org/) has tabulated a complete list of families of life, which total 9,650—on the whole a surprisingly small number. 10,000 barcodes, more or less, seems like a feasible goal. If we were able to assemble a complete database of DNA sequences at the family level, would it suffice to identify any eukaryote on Earth to the family level?

While the literature on species identification success of DNA barcodes comprises thousands of studies, only a few have tested their effectiveness at the level of higher

taxonomic units. In the seminal paper on DNA barcodes, *Hebert et al.* (*2003*) established that animal CO1 sequences can roughly assign taxa to phyla (96% success) or orders (100% success). However, their test was based on a neighbor joining tree-building approach, and it remained unknown if sequence data itself, i.e., percent identity among taxa, can be used in this way. Similarly, *Nagy et al.* (*2012*) showed that DNA barcoding in reptiles usually correctly assigned barcodes to species, genus and family. Their approach was phylogenetic: they tested whether including a sequence in tree building rendered the higher group non-monophyletic, which would imply failure. Finally, *Wilson et al.* (*2011*) provided a similar tree based test in sphingid moths, and established reliabilities of correct generic and subfamily taxonomic assignments between 74 and 90% using a liberal, and only 66–84% using a strict, tree-based criterion. These authors argued that tree-based methods perform better than sequence comparison methods, but that reliability, of course, depends on the library completeness.

Our project not only contributes original DNA barcode data for Central European spiders, but also works in synergy with the GGI towards a permanent preservation of genomic biodiversity: the formation of a collection of deeply frozen spider tissues and their DNA. We provide: (1) cryo-preserved tissues of reliably identified species of Central European spiders, and their vouchers photographed and deposited in public museums; (2) permanently frozen genomic DNA of these species; (3) publicly accessible DNA barcodes for these species (genetic sequence of cytochrome oxidase I—CO1) as public identification tool (*Hebert et al.*, *2003*) to facilitate organism identification, taxonomy, ecology and conservation.

In addition, this project addresses to what extent higher level taxonomic units can be reliably identified using barcodes of unknown spiders, and specifically asks what percent sequence identity in BLAST results is necessary to correctly identify unknown taxa to the Linnaean genus and/or family. Other methods for classification of higher-level taxonomies such as RDP (*Wang et al.*, *2007*), UTAX (*Edgar*, *2010*) and MEGAN (*Huson et al.*, *2007*) have primarily been developed for studies of microorganisms, using genetic markers for these groups, but less is known about using the CO1 barcoding gene in metazoans. We examine empirical data from Araneae barcode data to ask what is the percent sequence identity value above which 5% or less of higher level (genus/family) taxonomic identifications are incorrect and the extent to which frequency of correct identifications correlated with the number of taxa in this dataset, as would be expected given the dependence of BLAST on the reference database.

## MATERIALS & METHODS

### Specimen processing and imaging

We used automated and manual sampling methods for collecting spiders in the field in numerous localities in Slovenia and Switzerland. Faunistic and sampling details are published elsewhere (*Čandek et al.*, *2013*; see also *2015* corrigendum). Collected spiders were fixed in absolute ethanol immediately after being caught and the ethanol was replaced on the following day. Spiders were frozen at −80 °C, same day, or as soon as possible. In the

laboratory they were identified, labeled, photographed and processed for DNA extraction and sequencing (*Čandek et al.*, *2013*; see also *2015* corrigendum). Voucher specimens (voucher codes starting with 0078) are deposited at National Museum of Natural History, Smithsonian Institution (Washington D.C., USA), with duplicates (voucher codes starting with ARA) at Naturhistorisches Museum der Burgergemeinde Bern (Switzerland) and EZ LAB, ZRC SAZU (Ljubljana, Slovenia).

Voucher images are published along with their barcodes (see Table 1) at http://ezlab.zrc-sazu.si/dna. All original sequences generated by this project have been submitted to BOLD systems, and those that BOLD accepted were also submitted to GenBank (Table 1).

## Tissues

After specimen identification and processing, up to four legs (or in the case of very small individuals the whole prosoma) of a spider were removed and stored in fresh absolute ethanol in cryovials. Part of the tissue was used for DNA isolation while the other part remains permanently frozen at −80 °C at GGI facilities. The maintenance and use of these materials abides by the international legal standards and conventions of the biological genetic heritage (The Access and Benefit Sharing agreement as part of the 2010 Nagoya protocol).

## Molecular procedures

At Laboratories of Analytical Biology (National Museum of Natural History, Smithsonian Institution, hereafter LAB), specimens were extracted using the AutoGenPrep phenol-chloroform automated extractor (AutoGen). Samples were digested overnight in buffer containing proteinase-k before extraction. At EZ Lab, specimens were extracted using the Mag MAX$^{TM}$ Express magnetic particle processor Type 700 with DNA Multisample kit (Applied Biosystems, Foster City, CA, USA) following the manufacturer's protocols with modifications (*Vidergar, Toplak & Kuntner*, *2014*).

At EZ Lab PCR was carried out using mainly primers LCO1490 and HCO2198 (*Folmer et al.*, *1994*). Standard reaction volume was 35 µL containing 2.3 mM MgCl$_2$ (Promega), 0.15 mM each dNTP (Biotools), 0.4 µM of each primer, 0.2 µL 10 mg/mL BSA (Promega), 0.2 µL GoTaqFlexi polymerase (Promega) and 2 µL DNA. PCR cycling conditions were as follows: an initial denaturation step of 2 min at 94 °C followed by 35 cycles of 40 s at 94 °C, 1 min at 48 °–52 °C, 1 min at 72 °C, with final extension at 72 °C for 3 min. Additional primers were used for PCR for a few problematic specimens: dgLCO1490 and dgHCO2198 (*Meyer & Paulay*, *2005*) and the reverse primer Chelicerate-R2 (*Barrett & Hebert*, *2005*). Cycling parameters for difficult specimens were: 20 cycles of usual cycling protocol (above) followed by 15 cycles of 1.5 min at 94 °C, 1.5 min at 52 °C and 2 min at 72 °Cm version 5.6.6 (*Kearse et al.*, *2012*). EZ Lab PCR products were sent to be Sanger sequenced at Macrogen Inc. (Amsterdam, Netherlands), and the sequences were aligned, checked for sequencing errors and trimmed to match the barcode region in Geneious Pro version 5.6.6 (*Kearse et al.*, *2012*).

At LAB, PCR was carried out using the primer pair LCO1490 (*Folmer et al.*, *1994*) and Chelicerate-R2 (*Barrett & Hebert*, *2005*). A 10 µL reaction mix contained 2.5 mM MgCl$_2$

**Table 1** Original sequences this project submitted to BOLD and GenBank (only those on GenBank are also publically available on BOLD, for all others, see http://ezlab.zrc-sazu.si/dna/).

| Family | Genus | Species | Sample ID | Process ID | GenBank accession number | Voucher stored at | Collected in |
|---|---|---|---|---|---|---|---|
| Agelenidae | Agelena | labyrinthica | 00786574 | SPSLO002-12 | | MNH, SI | SVN |
| Agelenidae | Agelena | labyrinthica | ARA0239 | SPSLO369-13 | | EZ LAB | SVN |
| Agelenidae | Allagelena | gracilens | 00786557 | SPSLO001-12 | KX039062 | MNH, SI | SVN |
| Agelenidae | Coelotes | terrestris | 00786563 | SPSLO003-12 | KX039130 | MNH, SI | SVN |
| Agelenidae | Histopona | torpida | 00786599 | SPSLO004-12 | KX039207 | MNH, SI | SVN |
| Agelenidae | Histopona | torpida | ARA0063 | SPSLO339-13 | KX039208 | EZ LAB | SVN |
| Agelenidae | Inermocoelotes | anoplus | 00786586 | SPSLO005-12 | KX039220 | MNH, SI | SVN |
| Agelenidae | Inermocoelotes | anoplus | ARA0339 | SPSLO392-13 | KX039219 | EZ LAB | SVN |
| Agelenidae | Malthonica | silvestris | 00786304 | SPSLO283-13 | KX039239 | MNH, SI | SVN |
| Agelenidae | Malthonica | silvestris | ARA0427 | SPSLO468-13 | KX039238 | EZ LAB | SVN |
| Agelenidae | Tegenaria | atrica | 00786583 | SPSLO006-12 | KX039170 | MNH, SI | SVN |
| Agelenidae | Tegenaria | atrica | ARA0076 | SPSLO341-13 | KX039169 | EZ LAB | SVN |
| Amaurobiidae | Amaurobius | erberi | 00786571 | SPSLO007-12 | KX039070 | MNH, SI | SVN |
| Amaurobiidae | Amaurobius | erberi | ARA0120 | SPSLO347-13 | KX039069 | EZ LAB | SVN |
| Amaurobiidae | Amaurobius | fenestralis | 00786389 | SPSLO189-12 | KX039071 | MNH, SI | CHE |
| Amaurobiidae | Amaurobius | ferox | 00786307 | SPSLO284-13 | KX039072 | MNH, SI | SVN |
| Amaurobiidae | Amaurobius | jugorum | 00786585 | SPSLO008-12 | KX039073 | MNH, SI | SVN |
| Anyphaenidae | Anyphaena | accentuata | 00786584 | SPSLO009-12 | KX039076 | MNH, SI | SVN |
| Anyphaenidae | Anyphaena | sabina | 00786551 | SPSLO010-12 | KX039077 | MNH, SI | SVN |
| Araneidae | Aculepeira | ceropegia | 00786570 | SPSLO011-12 | KX039041 | MNH, SI | SVN |
| Araneidae | Aculepeira | ceropegia | ARA0355 | SPSLO405-13 | KX039040 | NMBE | CHE |
| Araneidae | Agalenatea | redii | 00786368 | SPSLO095-12 | KX039043 | MNH, SI | SVN |
| Araneidae | Agalenatea | redii | ARA0381 | SPSLO429-13 | KX039042 | EZ LAB | SVN |
| Araneidae | Araneus | alsine | 00786568 | SPSLO012-12 | KX039079 | MNH, SI | SVN |
| Araneidae | Araneus | angulatus | 00786552 | SPSLO013-12 | KX039081 | MNH, SI | SVN |
| Araneidae | Araneus | angulatus | ARA0001 | SPSLO326-13 | KX039080 | EZ LAB | SVN |
| Araneidae | Araneus | diadematus | 00786593 | SPSLO014-12 | KX039083 | MNH, SI | SVN |
| Araneidae | Araneus | diadematus | ARA0050 | SPSLO336-13 | KX039082 | EZ LAB | SVN |
| Araneidae | Araneus | marmoreus | 00786575 | SPSLO015-12 | KX039085 | MNH, SI | SVN |
| Araneidae | Araneus | marmoreus | ARA0030 | SPSLO329-13 | KX039084 | EZ LAB | SVN |
| Araneidae | Araneus | quadratus | 00786572 | SPSLO016-12 | KX039086 | MNH, SI | SVN |
| Araneidae | Araneus | quadratus | ARA0198 | SPSLO362-13 | KX039087 | NMBE | CHE |
| Araneidae | Araneus | sturmi | 00786561 | SPSLO017-12 | KX039089 | MNH, SI | SVN |
| Araneidae | Araneus | sturmi | ARA0108 | SPSLO345-13 | KX039088 | EZ LAB | SVN |
| Araneidae | Araniella | cucurbitina | 00786596 | SPSLO018-12 | KX039090 | MNH, SI | SVN |
| Araneidae | Araniella | opisthographa | ARA0393 | SPSLO440-13 | KX039091 | EZ LAB | SVN |
| Araneidae | Argiope | bruennichi | 00786589 | SPSLO019-12 | KX039093 | MNH, SI | SVN |
| Araneidae | Argiope | bruennichi | ARA0048 | SPSLO335-13 | KX039094 | EZ LAB | SVN |

*(continued on next page)*

**Table 1** (*continued*)

| Family | Genus | Species | Sample ID | Process ID | GenBank accession number | Voucher stored at | Collected in |
|---|---|---|---|---|---|---|---|
| Araneidae | Cercidia | prominens | 00786498 | SPSLO021-12 | KX039116 | MNH, SI | SVN |
| Araneidae | Cercidia | prominens | 00786577 | SPSLO020-12 | KX039114 | MNH, SI | SVN |
| Araneidae | Cercidia | prominens | ARA0356 | SPSLO406-13 | KX039115 | EZ LAB | SVN |
| Araneidae | Cyclosa | conica | 00786573 | SPSLO022-12 | KX039135 | MNH, SI | SVN |
| Araneidae | Cyclosa | conica | ARA0380 | SPSLO428-13 | KX039136 | NMBE | CHE |
| Araneidae | Gibbaranea | bituberculata | 00786579 | SPSLO023-12 | KX039186 | MNH, SI | SVN |
| Araneidae | Gibbaranea | bituberculata | ARA0350 | SPSLO400-13 | KX039187 | EZ LAB | SVN |
| Araneidae | Hypsosinga | albovittata | 00786323 | SPSLO191-12 | KX039211 | MNH, SI | CHE |
| Araneidae | Hypsosinga | pygmaea | 00786555 | SPSLO024-12 | KX039212 | MNH, SI | SVN |
| Araneidae | Hypsosinga | sanguinea | 00786314 | SPSLO285-13 | KX039213 | MNH, SI | SVN |
| Araneidae | Hypsosinga | sanguinea | ARA0370 | SPSLO419-13 | KX039214 | NMBE | CHE |
| Araneidae | Larinioides | sclopetarius | 00786382 | SPSLO096-12 | KX039222 | MNH, SI | SVN |
| Araneidae | Leviellus | thorelli | 00786591 | SPSLO025-12 | KX039229 | MNH, SI | SVN |
| Araneidae | Leviellus | thorelli | ARA0353 | SPSLO403-13 | KX039228 | EZ LAB | SVN |
| Araneidae | Mangora | acalypha | 00786590 | SPSLO026-12 | KX039242 | MNH, SI | SVN |
| Araneidae | Mangora | acalypha | ARA0107 | SPSLO344-13 | KX039240 | EZ LAB | SVN |
| Araneidae | Mangora | acalypha | ARA0357 | SPSLO407-13 | KX039241 | EZ LAB | SVN |
| Araneidae | Neoscona | adianta | 00786330 | SPSLO192-12 | KX039282 | MNH, SI | SVN |
| Araneidae | Nuctenea | umbratica | 00786594 | SPSLO027-12 | KX039293 | MNH, SI | SVN |
| Araneidae | Nuctenea | umbratica | ARA0387 | SPSLO435-13 | KX039292 | NMBE | CHE |
| Araneidae | Parazygiella | montana | 00786582 | SPSLO028-12 | KX039307 | MNH, SI | SVN |
| Araneidae | Parazygiella | montana | ARA0354 | SPSLO404-13 | KX039308 | NMBE | CHE |
| Araneidae | Singa | nitidula | 00786597 | SPSLO029-12 | KX039376 | MNH, SI | SVN |
| Araneidae | Stroemiellus | stroemi | ARA0169 | SPSLO358-13 | KX039383 | EZ LAB | SVN |
| Araneidae | Zilla | diodia | 00786481 | SPSLO097-12 | KX039446 | MNH, SI | SVN |
| Araneidae | Zilla | diodia | ARA0342 | SPSLO393-13 | KX039447 | EZ LAB | SVN |
| Araneidae | Zygiella | x-notata | 00786326 | SPSLO193-12 | KX039450 | MNH, SI | CHE |
| Atypidae | Atypus | piceus | 00786580 | SPSLO031-12 | KX039096 | MNH, SI | SVN |
| Atypidae | Atypus | piceus | ARA0174 | SPSLO359-13 | KX039097 | EZ LAB | SVN |
| Clubionidae | Clubiona | germanica | 00786566 | SPSLO032-12 | KX039122 | MNH, SI | SVN |
| Clubionidae | Clubiona | kulczynskii | 00786404 | SPSLO194-12 | KX039123 | MNH, SI | CHE |
| Clubionidae | Clubiona | neglecta | 00786558 | SPSLO033-12 | KX039124 | MNH, SI | SVN |
| Clubionidae | Clubiona | pseudoneglecta | 00786286 | SPSLO286-13 | KX039125 | MNH, SI | SVN |
| Clubionidae | Clubiona | reclusa | 00786378 | SPSLO195-12 | KX039126 | MNH, SI | CHE |
| Clubionidae | Clubiona | reclusa | ARA0371 | SPSLO420-13 | KX039127 | NMBE | CHE |
| Clubionidae | Clubiona | terrestris | 00786457 | SPSLO098-12 | KX039129 | MNH, SI | SVN |
| Clubionidae | Clubiona | terrestris | ARA0242 | SPSLO371-13 | KX039128 | EZ LAB | SVN |
| Corinnidae | Phrurolithus | minimus | 00786559 | SPSLO034-12 | KX039341 | MNH, SI | SVN |
| Dictynidae | Argenna | subnigra | 00786283 | SPSLO288-13 | | MNH, SI | SVN |
| Dictynidae | Cicurina | cicur | 00786548 | SPSLO035-12 | KX039121 | MNH, SI | SVN |
| Dictynidae | Dictyna | arundinacea | 00786369 | SPSLO196-12 | KX039140 | MNH, SI | CHE |
| Dictynidae | Dictyna | arundinacea | ARA0379 | SPSLO427-13 | KX039139 | NMBE | CHE |

**Table 1** (*continued*)

| Family | Genus | Species | Sample ID | Process ID | GenBank accession number | Voucher stored at | Collected in |
|---|---|---|---|---|---|---|---|
| Dictynidae | Dictyna | civica | 00786511 | SPSLO036-12 | KX039141 | MNH, SI | SVN |
| Dictynidae | Dictyna | uncinata | 00786345 | SPSLO197-12 | KX039143 | MNH, SI | SVN |
| Dictynidae | Dictyna | uncinata | ARA0423 | SPSLO466-13 | KX039142 | EZ LAB | SVN |
| Dictynidae | Lathys | humilis | 00786473 | SPSLO198-12 | KX039224 | MNH, SI | SVN |
| Dictynidae | Lathys | humilis | 00786550 | SPSLO037-12 | KX039223 | MNH, SI | SVN |
| Dysderidae | Dasumia | canestrinii | 00786581 | SPSLO038-12 | KX039137 | MNH, SI | SVN |
| Dysderidae | Dysdera | adriatica | 00786287 | SPSLO289-13 | KX039154 | MNH, SI | SVN |
| Dysderidae | Dysdera | adriatica | 00786296 | SPSLO290-13 | KX039155 | MNH, SI | SVN |
| Dysderidae | Dysdera | ninnii | ARA0244 | SPSLO373-13 | KX039156 | EZ LAB | SVN |
| Filistatidae | Filistata | insidiatrix | 00786560 | SPSLO040-12 | KX039181 | MNH, SI | SVN |
| Filistatidae | Filistata | insidiatrix | ARA0122 | SPSLO348-13 | KX039182 | EZ LAB | SVN |
| Gnaphosidae | Aphantaulax | cincta | 00786470 | SPSLO199-12 | KX039078 | MNH, SI | SVN |
| Gnaphosidae | Callilepis | schuszteri | 00786553 | SPSLO041-12 | KX039103 | MNH, SI | SVN |
| Gnaphosidae | Callilepis | schuszteri | ARA0333 | SPSLO386-13 | KX039102 | EZ LAB | SVN |
| Gnaphosidae | Drassodes | lapidosus | 00786505 | SPSLO099-12 | KX039150 | MNH, SI | SVN |
| Gnaphosidae | Drassodes | pubescens | 00786273 | SPSLO291-13 | KX039151 | MNH, SI | SVN |
| Gnaphosidae | Drassyllus | villicus | 00786556 | SPSLO042-12 | KX039153 | MNH, SI | SVN |
| Gnaphosidae | Drassyllus | villicus | ARA0337 | SPSLO390-13 | KX039152 | EZ LAB | SVN |
| Gnaphosidae | Gnaphosa | bicolor | 00786276 | SPSLO292-13 | KX039188 | MNH, SI | SVN |
| Gnaphosidae | Haplodrassus | silvestris | 00786578 | SPSLO043-12 | KX039196 | MNH, SI | SVN |
| Gnaphosidae | Micaria | aenea | 00786384 | SPSLO200-12 | KX039258 | MNH, SI | CHE |
| Gnaphosidae | Micaria | pulicaria | 00786274 | SPSLO293-13 | KX039259 | MNH, SI | SVN |
| Gnaphosidae | Nomisia | exornata | 00786564 | SPSLO044-12 | KX039291 | MNH, SI | SVN |
| Gnaphosidae | Phaeocedus | braccatus | 00786592 | SPSLO045-12 | KX039330 | MNH, SI | SVN |
| Gnaphosidae | Scotophaeus | scutulatus | 00786576 | SPSLO046-12 | KX039369 | MNH, SI | SVN |
| Gnaphosidae | Scotophaeus | scutulatus | ARA0082 | SPSLO343-13 | KX039370 | EZ LAB | SVN |
| Gnaphosidae | Trachyzelotes | pedestris | 00786279 | SPSLO294-13 | KX039419 | MNH, SI | SVN |
| Gnaphosidae | Zelotes | apricorum | 00786278 | SPSLO295-13 | KX039441 | MNH, SI | SVN |
| Gnaphosidae | Zelotes | latreillei | 00786540 | SPSLO047-12 | KX039443 | MNH, SI | SVN |
| Gnaphosidae | Zelotes | latreillei | ARA0191 | SPSLO360-13 | KX039442 | EZ LAB | SVN |
| Gnaphosidae | Zelotes | subterraneus | 00786588 | SPSLO048-12 | KX039445 | MNH, SI | CHE |
| Gnaphosidae | Zelotes | subterraneus | ARA0156 | SPSLO355-13 | KX039444 | NMBE | CHE |
| Hahniidae | Antistea | elegans | 00786405 | SPSLO201-12 | KX039075 | MNH, SI | CHE |
| Hahniidae | Antistea | elegans | ARA0384 | SPSLO432-13 | KX039074 | NMBE | CHE |
| Hahniidae | Hahnia | difficilis | 00786363 | SPSLO202-12 | KX039195 | MNH, SI | CHE |
| Hahniidae | Hahnia | difficilis | ARA0399 | SPSLO445-13 | KX039194 | NMBE | CHE |
| Linyphiidae | Agnyphantes | expunctus | 00786328 | SPSLO203-12 | KX039044 | MNH, SI | CHE |
| Linyphiidae | Agnyphantes | expunctus | ARA0429 | SPSLO470-13 | KX039045 | NMBE | CHE |
| Linyphiidae | Agyneta | affinis | 00786439 | SPSLO115-12 | KX039049 | MNH, SI | CHE |
| Linyphiidae | Agyneta | affinis | ARA0245 | SPSLO374-13 | KX039048 | NMBE | CHE |
| Linyphiidae | Agyneta | alpica | 00786443 | SPSLO116-12 | KX039050 | MNH, SI | CHE |
| Linyphiidae | Agyneta | cauta | 00786426 | SPSLO204-12 | KX039052 | MNH, SI | CHE |

| Family | Genus | Species | Sample ID | Process ID | GenBank accession number | Voucher stored at | Collected in |
|---|---|---|---|---|---|---|---|
| Linyphiidae | Agyneta | cauta | ARA0225 | SPSLO367-13 | KX039051 | NMBE | CHE |
| Linyphiidae | Agyneta | conigera | 00786448 | SPSLO100-12 | KX039053 | MNH, SI | CHE |
| Linyphiidae | Agyneta | fuscipalpa | 00786425 | SPSLO218-12 | | MNH, SI | CHE |
| Linyphiidae | Agyneta | fuscipalpa | ARA0268 | SPSLO378-13 | | NMBE | CHE |
| Linyphiidae | Agyneta | gulosa | 00786464 | SPSLO219-12 | KX039054 | MNH, SI | CHE |
| Linyphiidae | Agyneta | innotabilis | 00786393 | SPSLO220-12 | KX039055 | MNH, SI | CHE |
| Linyphiidae | Agyneta | orites | 00786419 | SPSLO221-12 | KX039057 | MNH, SI | CHE |
| Linyphiidae | Agyneta | orites | ARA0403 | SPSLO449-13 | KX039056 | NMBE | CHE |
| Linyphiidae | Agyneta | rurestris | 00786411 | SPSLO117-12 | KX039058 | MNH, SI | CHE |
| Linyphiidae | Agyneta | rurestris | ARA0419 | SPSLO462-13 | KX039059 | EZ LAB | SVN |
| Linyphiidae | Agyneta | saxatilis | 00786277 | SPSLO298-13 | KX039060 | MNH, SI | SVN |
| Linyphiidae | Agyneta | simplicitarsis | 00786295 | SPSLO299-13 | KX039061 | MNH, SI | SVN |
| Linyphiidae | Bolyphantes | alticeps | 00786465 | SPSLO205-12 | | MNH, SI | CHE |
| Linyphiidae | Bolyphantes | luteolus | 00786397 | SPSLO101-12 | KX039101 | MNH, SI | CHE |
| Linyphiidae | Bolyphantes | luteolus | ARA0214 | SPSLO366-13 | KX039100 | NMBE | CHE |
| Linyphiidae | Caracladus | avicula | 00786474 | SPSLO206-12 | KX039104 | MNH, SI | CHE |
| Linyphiidae | Caracladus | avicula | ARA0231 | SPSLO368-13 | KX039105 | NMBE | CHE |
| Linyphiidae | Caracladus | zamoniensis | 00786441 | SPSLO102-12 | KX039106 | MNH, SI | CHE |
| Linyphiidae | Centromerus | pabulator | 00786451 | SPSLO207-12 | KX039108 | MNH, SI | CHE |
| Linyphiidae | Centromerus | pabulator | ARA0421 | SPSLO464-13 | KX039107 | NMBE | CHE |
| Linyphiidae | Centromerus | subalpinus | 00786412 | SPSLO208-12 | KX039110 | MNH, SI | CHE |
| Linyphiidae | Centromerus | subalpinus | ARA0250 | SPSLO375-13 | KX039109 | NMBE | CHE |
| Linyphiidae | Ceratinella | brevipes | 00786317 | SPSLO234-12 | KX039112 | MNH, SI | CHE |
| Linyphiidae | Ceratinella | brevipes | 00786450 | SPSLO103-12 | KX039113 | MNH, SI | CHE |
| Linyphiidae | Ceratinella | brevipes | ARA0363 | SPSLO413-13 | KX039111 | NMBE | CHE |
| Linyphiidae | Diplocephalus | crassilobus | 00786294 | SPSLO296-13 | KX039144 | MNH, SI | SVN |
| Linyphiidae | Diplocephalus | latifrons | 00786461 | SPSLO209-12 | KX039145 | MNH, SI | CHE |
| Linyphiidae | Diplostyla | concolor | 00786533 | SPSLO049-12 | KX039146 | MNH, SI | SVN |
| Linyphiidae | Drapetisca | socialis | 00786587 | SPSLO050-12 | KX039149 | MNH, SI | SVN |
| Linyphiidae | Drapetisca | socialis | ARA0405 | SPSLO451-13 | KX039148 | EZ LAB | SVN |
| Linyphiidae | Entelecara | acuminata | 00786460 | SPSLO210-12 | KX039164 | MNH, SI | CHE |
| Linyphiidae | Erigone | atra | ARA0257 | SPSLO377-13 | KX039171 | NMBE | CHE |
| Linyphiidae | Erigone | dentipalpis | ARA0256 | SPSLO376-13 | KX039172 | NMBE | CHE |
| Linyphiidae | Erigone | remota | 00786416 | SPSLO107-12 | KX039174 | MNH, SI | CHE |
| Linyphiidae | Erigonella | ignobilis | ARA0164 | SPSLO357-13 | KX039173 | NMBE | CHE |
| Linyphiidae | Floronia | bucculenta | 00786545 | SPSLO051-12 | KX039183 | MNH, SI | SVN |
| Linyphiidae | Frontinellina | frutetorum | 00786567 | SPSLO052-12 | KX039184 | MNH, SI | SVN |
| Linyphiidae | Frontinellina | frutetorum | ARA0441 | SPSLO480-13 | KX039185 | EZ LAB | SVN |
| Linyphiidae | Gonatium | hilare | 00786565 | SPSLO053-12 | KX039189 | MNH, SI | SVN |
| Linyphiidae | Gonatium | rubellum | 00786318 | SPSLO212-12 | KX039191 | MNH, SI | CHE |
| Linyphiidae | Gonatium | rubellum | ARA0386 | SPSLO434-13 | KX039190 | NMBE | CHE |
| Linyphiidae | Gonatium | rubens | 00786331 | SPSLO213-12 | KX039192 | MNH, SI | CHE |

| Family | Genus | Species | Sample ID | Process ID | GenBank accession number | Voucher stored at | Collected in |
|---|---|---|---|---|---|---|---|
| Linyphiidae | Gonatium | rubens | ARA0358 | SPSLO408-13 | KX039193 | NMBE | CHE |
| Linyphiidae | Improphantes | nitidus | 00786449 | SPSLO109-12 | KX039218 | MNH, SI | CHE |
| Linyphiidae | Incestophantes | frigidus | ARA0211 | SPSLO364-13 | | NMBE | CHE |
| Linyphiidae | Kaestneria | dorsalis | 00786598 | SPSLO054-12 | KX039221 | MNH, SI | SVN |
| Linyphiidae | Lepthyphantes | leprosus | 00786342 | SPSLO214-12 | KX039225 | MNH, SI | SVN |
| Linyphiidae | Lepthyphantes | nodifer | ARA0433 | SPSLO473-13 | KX039226 | NMBE | CHE |
| Linyphiidae | Linyphia | hortensis | 00786526 | SPSLO112-12 | KX039230 | MNH, SI | SVN |
| Linyphiidae | Linyphia | hortensis | ARA0397 | SPSLO443-13 | KX039231 | NMBE | CHE |
| Linyphiidae | Linyphia | triangularis | 00786547 | SPSLO056-12 | KX039232 | MNH, SI | SVN |
| Linyphiidae | Linyphia | triangularis | ARA0004 | SPSLO327-13 | KX039233 | EZ LAB | SVN |
| Linyphiidae | Macrargus | rufus | ARA0213 | SPSLO365-13 | KX039237 | NMBE | CHE |
| Linyphiidae | Mansuphantes | fragilis | 00786415 | SPSLO114-12 | KX039243 | MNH, SI | CHE |
| Linyphiidae | Mansuphantes | fragilis | ARA0276 | SPSLO380-13 | KX039244 | NMBE | CHE |
| Linyphiidae | Maso | sundevalli | 00786400 | SPSLO216-12 | KX039248 | MNH, SI | CHE |
| Linyphiidae | Maso | sundevalli | ARA0360 | SPSLO410-13 | KX039247 | NMBE | CHE |
| Linyphiidae | Megalepthyphantes | collinus | 00786569 | SPSLO057-12 | KX039249 | MNH, SI | SVN |
| Linyphiidae | Mermessus | trilobatus | 00786395 | SPSLO118-12 | KX039250 | MNH, SI | SVN |
| Linyphiidae | Metopobactrus | prominulus | 00786437 | SPSLO119-12 | KX039257 | MNH, SI | CHE |
| Linyphiidae | Micrargus | alpinus | ARA0270 | SPSLO379-13 | KX039260 | NMBE | CHE |
| Linyphiidae | Micrargus | herbigradus | 00786466 | SPSLO223-12 | KX039261 | MNH, SI | CHE |
| Linyphiidae | Microctenonyx | subitaneus | 00786463 | SPSLO224-12 | KX039262 | MNH, SI | CHE |
| Linyphiidae | Microlinyphia | impigra | 00786350 | SPSLO228-12 | KX039263 | MNH, SI | CHE |
| Linyphiidae | Microlinyphia | impigra | ARA0369 | SPSLO418-13 | KX039264 | NMBE | CHE |
| Linyphiidae | Microlinyphia | pusilla | 00786417 | SPSLO225-12 | KX039265 | MNH, SI | CHE |
| Linyphiidae | Minicia | marginella | 00786371 | SPSLO120-12 | KX039267 | MNH, SI | SVN |
| Linyphiidae | Minicia | marginella | ARA0410 | SPSLO455-13 | KX039268 | NMBE | CHE |
| Linyphiidae | Minyriolus | pusillus | ARA0285 | SPSLO382-13 | KX039269 | NMBE | CHE |
| Linyphiidae | Mughiphantes | cornutus | ARA0372 | SPSLO421-13 | KX039272 | NMBE | CHE |
| Linyphiidae | Mughiphantes | mughi | 00786319 | SPSLO227-12 | KX039274 | MNH, SI | CHE |
| Linyphiidae | Mughiphantes | mughi | 00786322 | SPSLO217-12 | KX039275 | MNH, SI | CHE |
| Linyphiidae | Mughiphantes | mughi | ARA0361 | SPSLO411-13 | KX039273 | NMBE | CHE |
| Linyphiidae | Mughiphantes | mughi | ARA0411 | SPSLO456-13 | KX039276 | NMBE | CHE |
| Linyphiidae | Nematogmus | sanguinolentus | 00786490 | SPSLO162-12 | KX039279 | MNH, SI | SVN |
| Linyphiidae | Nematogmus | sanguinolentus | ARA0359 | SPSLO409-13 | KX039278 | NMBE | CHE |
| Linyphiidae | Neriene | clathrata | ARA0352 | SPSLO402-13 | KX039287 | EZ LAB | SVN |
| Linyphiidae | Neriene | furtiva | 00786471 | SPSLO229-12 | KX039289 | MNH, SI | SVN |
| Linyphiidae | Neriene | furtiva | ARA0145 | SPSLO353-13 | KX039288 | EZ LAB | SVN |
| Linyphiidae | Neriene | radiata | ARA0152 | SPSLO354-13 | KX039290 | NMBE | CHE |
| Linyphiidae | Obscuriphantes | obscurus | 00786354 | SPSLO231-12 | KX039295 | MNH, SI | CHE |
| Linyphiidae | Obscuriphantes | obscurus | ARA0407 | SPSLO453-13 | KX039294 | NMBE | CHE |
| Linyphiidae | Oedothorax | gibbifer | 00786396 | SPSLO232-12 | KX039296 | MNH, SI | CHE |
| Linyphiidae | Oryphantes | angulatus | ARA0398 | SPSLO444-13 | | NMBE | CHE |

| Family | Genus | Species | Sample ID | Process ID | GenBank accession number | Voucher stored at | Collected in |
|---|---|---|---|---|---|---|---|
| Linyphiidae | Ostearius | melanopygius | 00786339 | SPSLO122-12 | KX039297 | MNH, SI | SVN |
| Linyphiidae | Palliduphantes | pallidus | 00786341 | SPSLO233-12 | KX039302 | MNH, SI | CHE |
| Linyphiidae | Panamomops | tauricornis | ARA0375 | SPSLO424-13 | KX039303 | NMBE | CHE |
| Linyphiidae | Pityohyphantes | phrygianus | 00786316 | SPSLO236-12 | KX039351 | MNH, SI | CHE |
| Linyphiidae | Pityohyphantes | phrygianus | ARA0347 | SPSLO397-13 | KX039352 | NMBE | CHE |
| Linyphiidae | Pocadicnemis | juncea | 00786421 | SPSLO237-12 | KX039354 | MNH, SI | CHE |
| Linyphiidae | Pocadicnemis | juncea | ARA0409 | SPSLO454-13 | KX039355 | NMBE | CHE |
| Linyphiidae | Pocadicnemis | pumila | 00786422 | SPSLO238-12 | KX039356 | MNH, SI | CHE |
| Linyphiidae | Porrhomma | pallidum | 00786410 | SPSLO239-12 | KX039357 | MNH, SI | CHE |
| Linyphiidae | Porrhomma | pygmaeum | 00786292 | SPSLO301-13 | | MNH, SI | SVN |
| Linyphiidae | Scotinotylus | alpigena | 00786444 | SPSLO125-12 | KX039367 | MNH, SI | CHE |
| Linyphiidae | Scotinotylus | alpigena | ARA0163 | SPSLO356-13 | KX039366 | NMBE | CHE |
| Linyphiidae | Scotinotylus | clavatus | 00786420 | SPSLO240-12 | KX039368 | MNH, SI | CHE |
| Linyphiidae | Silometopus | elegans | 00786409 | SPSLO126-12 | KX039373 | MNH, SI | CHE |
| Linyphiidae | Tapinocyba | affinis | 00786406 | SPSLO127-12 | KX039387 | MNH, SI | CHE |
| Linyphiidae | Tapinocyba | affinis | ARA0362 | SPSLO412-13 | KX039386 | NMBE | CHE |
| Linyphiidae | Tenuiphantes | alacris | 00786343 | SPSLO241-12 | KX039389 | MNH, SI | CHE |
| Linyphiidae | Tenuiphantes | alacris | ARA0420 | SPSLO463-13 | KX039388 | NMBE | CHE |
| Linyphiidae | Tenuiphantes | cristatus | 00786305 | SPSLO302-13 | KX039390 | MNH, SI | CHE |
| Linyphiidae | Tenuiphantes | cristatus | ARA0418 | SPSLO461-13 | KX039391 | NMBE | CHE |
| Linyphiidae | Tenuiphantes | flavipes | 00786528 | SPSLO060-12 | KX039392 | MNH, SI | SVN |
| Linyphiidae | Tenuiphantes | flavipes | ARA0336 | SPSLO389-13 | KX039393 | NMBE | CHE |
| Linyphiidae | Tenuiphantes | jacksoni | 00786356 | SPSLO242-12 | | MNH, SI | CHE |
| Linyphiidae | Tenuiphantes | jacksoni | 00786430 | SPSLO128-12 | | MNH, SI | CHE |
| Linyphiidae | Tenuiphantes | jacksoni | ARA0435 | SPSLO475-13 | | NMBE | CHE |
| Linyphiidae | Tenuiphantes | jacksonoides | ARA0374 | SPSLO423-13 | KX039394 | NMBE | CHE |
| Linyphiidae | Tenuiphantes | mengei | 00786301 | SPSLO300-13 | KX039396 | MNH, SI | CHE |
| Linyphiidae | Tenuiphantes | mengei | 00786413 | SPSLO243-12 | KX039397 | MNH, SI | CHE |
| Linyphiidae | Tenuiphantes | mengei | ARA0415 | SPSLO459-13 | KX039395 | NMBE | CHE |
| Linyphiidae | Tenuiphantes | tenebricola | 00786418 | SPSLO244-12 | KX039398 | MNH, SI | CHE |
| Linyphiidae | Tenuiphantes | tenebricola | ARA0414 | SPSLO458-13 | KX039399 | NMBE | CHE |
| Linyphiidae | Tenuiphantes | tenuis | 00786383 | SPSLO129-12 | | MNH, SI | SVN |
| Linyphiidae | Tiso | aestivus | ARA0422 | SPSLO465-13 | KX039413 | NMBE | CHE |
| Linyphiidae | Tiso | vagans | 00786351 | SPSLO246-12 | KX039414 | MNH, SI | CHE |
| Linyphiidae | Tiso | vagans | ARA0401 | SPSLO447-13 | KX039415 | NMBE | CHE |
| Linyphiidae | Walckenaeria | antica | 00786429 | SPSLO130-12 | KX039421 | MNH, SI | CHE |
| Linyphiidae | Walckenaeria | furcillata | 00786431 | SPSLO131-12 | KX039422 | MNH, SI | CHE |
| Liocranidae | Agroeca | brunnea | 00786320 | SPSLO247-12 | KX039046 | MNH, SI | SVN |
| Liocranidae | Agroeca | brunnea | ARA0392 | SPSLO439-13 | KX039047 | EZ LAB | SVN |
| Liocranidae | Liocranum | rupicola | 00786516 | SPSLO061-12 | KX039234 | MNH, SI | SVN |
| Liphistiidae | Liphistius | sp | ARA0240 | SPSLO482-15 | KX039235 | EZ LAB | MYS |
| Lycosidae | Alopecosa | accentuata | 00786365 | SPSLO248-12 | | MNH, SI | CHE |

**Table 1** (*continued*)

| Family | Genus | Species | Sample ID | Process ID | GenBank accession number | Voucher stored at | Collected in |
|--------|-------|---------|-----------|------------|--------------------------|-------------------|--------------|
| Lycosidae | Alopecosa | pulverulenta | 00786527 | SPSLO063-12 | KX039064 | MNH, SI | SVN |
| Lycosidae | Alopecosa | pulverulenta | ARA0349 | SPSLO399-13 | KX039063 | NMBE | CHE |
| Lycosidae | Alopecosa | sulzeri | 00786452 | SPSLO249-12 | KX039065 | MNH, SI | SVN |
| Lycosidae | Alopecosa | taeniata | 00786538 | SPSLO062-12 | KX039066 | MNH, SI | CHE |
| Lycosidae | Alopecosa | trabalis | 00786509 | SPSLO064-12 | KX039067 | MNH, SI | SVN |
| Lycosidae | Alopecosa | trabalis | ARA0438 | SPSLO478-13 | KX039068 | EZ LAB | SVN |
| Lycosidae | Arctosa | fulvolineata | 00786336 | SPSLO250-12 | | MNH, SI | SVN |
| Lycosidae | Arctosa | lutetiana | 00786407 | SPSLO132-12 | | MNH, SI | SVN |
| Lycosidae | Arctosa | maculata | 00786312 | SPSLO305-13 | KX039092 | MNH, SI | SVN |
| Lycosidae | Aulonia | albimana | 00786524 | SPSLO133-12 | KX039099 | MNH, SI | SVN |
| Lycosidae | Aulonia | albimana | ARA0338 | SPSLO391-13 | KX039098 | EZ LAB | SVN |
| Lycosidae | Hogna | radiata | 00786502 | SPSLO065-12 | KX039210 | MNH, SI | SVN |
| Lycosidae | Hogna | radiata | ARA0368 | SPSLO417-13 | KX039209 | EZ LAB | SVN |
| Lycosidae | Pardosa | agrestis | 00786385 | SPSLO134-12 | KX039309 | MNH, SI | SVN |
| Lycosidae | Pardosa | amentata | 00786337 | SPSLO251-12 | KX039311 | MNH, SI | SVN |
| Lycosidae | Pardosa | amentata | ARA0413 | SPSLO457-13 | KX039310 | NMBE | CHE |
| Lycosidae | Pardosa | bifasciata | 00786453 | SPSLO252-12 | KX039312 | MNH, SI | SVN |
| Lycosidae | Pardosa | blanda | 00786358 | SPSLO253-12 | KX039314 | MNH, SI | CHE |
| Lycosidae | Pardosa | blanda | ARA0345 | SPSLO396-13 | KX039313 | NMBE | CHE |
| Lycosidae | Pardosa | cf. lugubris | 00786529 | SPSLO066-12 | KX039316 | MNH, SI | CHE |
| Lycosidae | Pardosa | cf. lugubris | ARA0065 | SPSLO340-13 | KX039315 | EZ LAB | SVN |
| Lycosidae | Pardosa | ferruginea | 00786309 | SPSLO306-13 | KX039317 | MNH, SI | CHE |
| Lycosidae | Pardosa | hortensis | 00786289 | SPSLO307-13 | KX039318 | MNH, SI | SVN |
| Lycosidae | Pardosa | oreophila | 00786310 | SPSLO308-13 | KX039319 | MNH, SI | CHE |
| Lycosidae | Pardosa | oreophila | 00786321 | SPSLO254-12 | KX039320 | MNH, SI | CHE |
| Lycosidae | Pardosa | oreophila | ARA0348 | SPSLO398-13 | KX039321 | NMBE | CHE |
| Lycosidae | Pardosa | palustris | 00786514 | SPSLO067-12 | KX039323 | MNH, SI | SVN |
| Lycosidae | Pardosa | palustris | ARA0406 | SPSLO452-13 | KX039322 | NMBE | CHE |
| Lycosidae | Pardosa | proxima | 00786311 | SPSLO309-13 | KX039324 | MNH, SI | SVN |
| Lycosidae | Pardosa | riparia | 00786315 | SPSLO310-13 | KX039326 | MNH, SI | SVN |
| Lycosidae | Pardosa | riparia | ARA0243 | SPSLO372-13 | KX039325 | NMBE | CHE |
| Lycosidae | Pirata | piraticus | 00786375 | SPSLO255-12 | KX039346 | MNH, SI | CHE |
| Lycosidae | Pirata | piraticus | ARA0430 | SPSLO471-13 | KX039347 | NMBE | CHE |
| Lycosidae | Piratula | hygrophila | 00786388 | SPSLO135-12 | KX039348 | MNH, SI | SVN |
| Lycosidae | Piratula | knorri | 00786402 | SPSLO136-12 | | MNH, SI | SVN |
| Lycosidae | Trochosa | spinipalpis | 00786344 | SPSLO137-12 | | MNH, SI | SVN |
| Lycosidae | Trochosa | spinipalpis | ARA0388 | SPSLO436-13 | | EZ LAB | SVN |
| Lycosidae | Xerolycosa | nemoralis | 00786541 | SPSLO068-12 | KX039424 | MNH, SI | CHE |
| Lycosidae | Xerolycosa | nemoralis | ARA0335 | SPSLO388-13 | KX039423 | NMBE | CHE |
| Mimetidae | Ero | furcata | 00786390 | SPSLO256-12 | KX039175 | MNH, SI | CHE |
| Miturgidae | Cheiracanthium | erraticum | 00786367 | SPSLO138-12 | KX039117 | MNH, SI | SVN |
| Miturgidae | Cheiracanthium | mildei | 00786355 | SPSLO139-12 | KX039118 | MNH, SI | SVN |

**Table 1** (*continued*)

| Family | Genus | Species | Sample ID | Process ID | GenBank accession number | Voucher stored at | Collected in |
|---|---|---|---|---|---|---|---|
| Miturgidae | Cheiracanthium | punctorium | 00786519 | SPSLO140-12 | KX039120 | MNH, SI | SVN |
| Miturgidae | Cheiracanthium | punctorium | ARA0056 | SPSLO337-13 | KX039119 | EZ LAB | SVN |
| Nemesiidae | Nemesia | pannonica | 00786333 | SPSLO311-13 | KX039280 | MNH, SI | SVN |
| Philodromidae | Philodromus | albidus | 00786272 | SPSLO312-13 | KX039332 | MNH, SI | SVN |
| Philodromidae | Philodromus | aureolus | 00786539 | SPSLO069-12 | KX039333 | MNH, SI | SVN |
| Philodromidae | Philodromus | cespitum | 00786513 | SPSLO070-12 | KX039335 | MNH, SI | CHE |
| Philodromidae | Philodromus | cespitum | ARA0400 | SPSLO446-13 | KX039334 | EZ LAB | SVN |
| Philodromidae | Philodromus | dispar | 00786492 | SPSLO142-12 | KX039336 | MNH, SI | SVN |
| Philodromidae | Philodromus | praedatus | 00786500 | SPSLO071-12 | KX039338 | MNH, SI | SVN |
| Philodromidae | Philodromus | praedatus | ARA0404 | SPSLO450-13 | KX039337 | NMBE | CHE |
| Philodromidae | Philodromus | pulchellus | 00786475 | SPSLO072-12 | KX039339 | MNH, SI | SVN |
| Philodromidae | Philodromus | pulchellus | ARA0344 | SPSLO395-13 | KX039340 | EZ LAB | SVN |
| Philodromidae | Philodromus | vagulus | 00786366 | SPSLO257-12 | | MNH, SI | CHE |
| Philodromidae | Philodromus | vagulus | ARA0351 | SPSLO401-13 | | NMBE | CHE |
| Philodromidae | Thanatus | formicinus | 00786530 | SPSLO073-12 | KX039403 | MNH, SI | SVN |
| Philodromidae | Tibellus | macellus | 00786493 | SPSLO074-12 | KX039412 | MNH, SI | SVN |
| Philodromidae | Tibellus | macellus | ARA0334 | SPSLO387-13 | KX039411 | EZ LAB | SVN |
| Pholcidae | Psilochorus | simoni | 00786501 | SPSLO076-12 | KX039359 | MNH, SI | SVN |
| Pisauridae | Pisaura | mirabilis | 00786487 | SPSLO144-12 | KX039349 | MNH, SI | SVN |
| Pisauridae | Pisaura | mirabilis | ARA0383 | SPSLO431-13 | KX039350 | NMBE | CHE |
| Salticidae | Evarcha | arcuata | 00786332 | SPSLO259-12 | KX039177 | MNH, SI | CHE |
| Salticidae | Evarcha | arcuata | ARA0062 | SPSLO338-13 | KX039178 | EZ LAB | SVN |
| Salticidae | Evarcha | falcata | 00786408 | SPSLO145-12 | | MNH, SI | SVN |
| Salticidae | Evarcha | falcata | ARA0037 | SPSLO331-13 | KX039179 | EZ LAB | SVN |
| Salticidae | Evarcha | jucunda | 00786503 | SPSLO077-12 | KX039180 | MNH, SI | SVN |
| Salticidae | Evarcha | michailovi | 00786313 | SPSLO313-13 | | MNH, SI | SVN |
| Salticidae | Evarcha | michailovi | 00786458 | SPSLO260-12 | | MNH, SI | SVN |
| Salticidae | Evarcha | michailovi | ARA0436 | SPSLO476-13 | | EZ LAB | SVN |
| Salticidae | Hasarius | adansoni | 00786348 | SPSLO261-12 | KX039197 | MNH, SI | SVN |
| Salticidae | Heliophanus | aeneus | 00786293 | SPSLO314-13 | | MNH, SI | SVN |
| Salticidae | Heliophanus | auratus | 00786282 | SPSLO315-13 | KX039198 | MNH, SI | SVN |
| Salticidae | Heliophanus | cupreus | 00786518 | SPSLO146-12 | KX039199 | MNH, SI | SVN |
| Salticidae | Heliophanus | cupreus | ARA0382 | SPSLO430-13 | KX039200 | NMBE | CHE |
| Salticidae | Heliophanus | flavipes | 00786510 | SPSLO147-12 | KX039202 | MNH, SI | SVN |
| Salticidae | Heliophanus | flavipes | ARA0396 | SPSLO442-13 | KX039201 | EZ LAB | SVN |
| Salticidae | Heliophanus | kochii | 00786495 | SPSLO078-12 | KX039203 | MNH, SI | SVN |
| Salticidae | Icius | subinermis | 00786381 | SPSLO148-12 | KX039217 | MNH, SI | SVN |
| Salticidae | Leptorchestes | berolinensis | 00786512 | SPSLO079-12 | KX039227 | MNH, SI | SVN |
| Salticidae | Macaroeris | nidicolens | 00786338 | SPSLO262-12 | KX039236 | MNH, SI | SVN |
| Salticidae | Marpissa | muscosa | 00786523 | SPSLO080-12 | KX039245 | MNH, SI | SVN |
| Salticidae | Marpissa | nivoyi | 00786496 | SPSLO081-12 | KX039246 | MNH, SI | SVN |
| Salticidae | Myrmarachne | formicaria | 00786432 | SPSLO149-12 | KX039277 | MNH, SI | SVN |

**Table 1** (*continued*)

| Family | Genus | Species | Sample ID | Process ID | GenBank accession number | Voucher stored at | Collected in |
|--------|-------|---------|-----------|------------|--------------------------|-------------------|--------------|
| Salticidae | Neon | reticulatus | 00786370 | SPSLO150-12 | KX039281 | MNH, SI | SVN |
| Salticidae | Pellenes | seriatus | 00786462 | SPSLO263-12 | KX039329 | MNH, SI | SVN |
| Salticidae | Pellenes | seriatus | 00786504 | SPSLO082-12 | KX039327 | MNH, SI | SVN |
| Salticidae | Pellenes | seriatus | ARA0439 | SPSLO479-13 | KX039328 | EZ LAB | SVN |
| Salticidae | Philaeus | chrysops | 00786472 | SPSLO264-12 | KX039331 | MNH, SI | SVN |
| Salticidae | Pseudeuophrys | lanigera | 00786280 | SPSLO316-13 | KX039358 | MNH, SI | SVN |
| Salticidae | Saitis | barbipes | 00786507 | SPSLO083-12 | KX039363 | MNH, SI | SVN |
| Salticidae | Salticus | scenicus | 00786362 | SPSLO265-12 | KX039364 | MNH, SI | CHE |
| Salticidae | Sibianor | aurocinctus | 00786377 | SPSLO266-12 | | MNH, SI | CHE |
| Salticidae | Sibianor | aurocinctus | ARA0385 | SPSLO433-13 | | NMBE | CHE |
| Salticidae | Sitticus | rupicola | 00786525 | SPSLO084-12 | KX039377 | MNH, SI | CHE |
| Salticidae | Sitticus | rupicola | ARA0378 | SPSLO426-13 | KX039378 | NMBE | CHE |
| Scytodidae | Scytodes | thoracica | 00786521 | SPSLO085-12 | KX039371 | MNH, SI | SVN |
| Segestriidae | Segestria | senoculata | 00786281 | SPSLO317-13 | KX039372 | MNH, SI | SVN |
| Sparassidae | Micrommata | virescens | 00786497 | SPSLO086-12 | | MNH, SI | SVN |
| Sparassidae | Micrommata | virescens | ARA0365 | SPSLO414-13 | KX039266 | NMBE | CHE |
| Tetragnathidae | Metellina | mengei | 00786536 | SPSLO087-12 | KX039251 | MNH, SI | CHE |
| Tetragnathidae | Metellina | mengei | ARA0373 | SPSLO422-13 | KX039252 | NMBE | CHE |
| Tetragnathidae | Metellina | merianae | 00786298 | SPSLO318-13 | KX039253 | MNH, SI | CHE |
| Tetragnathidae | Metellina | merianae | ARA0394 | SPSLO441-13 | KX039254 | EZ LAB | SVN |
| Tetragnathidae | Metellina | segmentata | 00786357 | SPSLO152-12 | KX039255 | MNH, SI | SVN |
| Tetragnathidae | Metellina | segmentata | ARA0431 | SPSLO472-13 | KX039256 | EZ LAB | SVN |
| Tetragnathidae | Pachygnatha | degeeri | 00786399 | SPSLO153-12 | KX039300 | MNH, SI | SVN |
| Tetragnathidae | Tetragnatha | nigrita | 00786534 | SPSLO088-12 | KX039400 | MNH, SI | SVN |
| Tetragnathidae | Tetragnatha | nigrita | ARA0041 | SPSLO332-13 | KX039401 | EZ LAB | SVN |
| Tetragnathidae | Tetragnatha | pinicola | 00786361 | SPSLO267-12 | KX039402 | MNH, SI | CHE |
| Tetragnathidae | Tetragnatha | pinicola | 00786520 | SPSLO155-12 | | MNH, SI | SVN |
| Theridiidae | Asagena | phalerata | 00786346 | SPSLO156-12 | KX039095 | MNH, SI | SVN |
| Theridiidae | Crustulina | guttata | 00786454 | SPSLO268-12 | KX039132 | MNH, SI | SVN |
| Theridiidae | Crustulina | guttata | ARA0437 | SPSLO477-13 | KX039131 | EZ LAB | SVN |
| Theridiidae | Crustulina | scabripes | 00786479 | SPSLO089-12 | KX039134 | MNH, SI | SVN |
| Theridiidae | Crustulina | scabripes | ARA0137 | SPSLO352-13 | KX039133 | EZ LAB | SVN |
| Theridiidae | Dipoena | melanogaster | 00786506 | SPSLO090-12 | KX039147 | MNH, SI | SVN |
| Theridiidae | Enoplognatha | afrodite | 00786532 | SPSLO157-12 | KX039160 | MNH, SI | SVN |
| Theridiidae | Enoplognatha | afrodite | ARA0135 | SPSLO350-13 | KX039159 | EZ LAB | SVN |
| Theridiidae | Enoplognatha | latimana | 00786329 | SPSLO269-12 | KX039161 | MNH, SI | CHE |
| Theridiidae | Enoplognatha | ovata | 00786515 | SPSLO158-12 | KX039163 | MNH, SI | SVN |
| Theridiidae | Enoplognatha | ovata | ARA0367 | SPSLO416-13 | KX039162 | NMBE | CHE |
| Theridiidae | Episinus | angulatus | 00786386 | SPSLO159-12 | KX039165 | MNH, SI | SVN |
| Theridiidae | Episinus | maculipes | 00786488 | SPSLO160-12 | KX039166 | MNH, SI | SVN |
| Theridiidae | Episinus | truncatus | 00786327 | SPSLO270-12 | KX039168 | MNH, SI | CHE |
| Theridiidae | Episinus | truncatus | ARA0132 | SPSLO349-13 | KX039167 | EZ LAB | SVN |

**Table 1** (*continued*)

| Family | Genus | Species | Sample ID | Process ID | GenBank accession number | Voucher stored at | Collected in |
|---|---|---|---|---|---|---|---|
| Theridiidae | Euryopis | flavomaculata | 00786468 | SPSLO271-12 | KX039176 | MNH, SI | SVN |
| Theridiidae | Heterotheridion | nigrovariegatum | 00786482 | SPSLO161-12 | KX039206 | MNH, SI | SVN |
| Theridiidae | Heterotheridion | nigrovariegatum | ARA0343 | SPSLO394-13 | KX039205 | EZ LAB | SVN |
| Theridiidae | Neottiura | bimaculata | 00786445 | SPSLO163-12 | KX039284 | MNH, SI | SVN |
| Theridiidae | Neottiura | bimaculata | ARA0366 | SPSLO415-13 | KX039283 | NMBE | CHE |
| Theridiidae | Neottiura | herbigrada | 00786467 | SPSLO272-12 | KX039285 | MNH, SI | SVN |
| Theridiidae | Neottiura | suaveolens | 00786427 | SPSLO164-12 | KX039286 | MNH, SI | SVN |
| Theridiidae | Paidiscura | pallens | 00786288 | SPSLO319-13 | KX039301 | MNH, SI | SVN |
| Theridiidae | Parasteatoda | lunata | 00786476 | SPSLO165-12 | KX039304 | MNH, SI | SVN |
| Theridiidae | Parasteatoda | tepidariorum | 00786531 | SPSLO091-12 | KX039305 | MNH, SI | SVN |
| Theridiidae | Parasteatoda | tepidariorum | ARA0329 | SPSLO384-13 | KX039306 | EZ LAB | SVN |
| Theridiidae | Phylloneta | impressa | 00786401 | SPSLO273-12 | KX039342 | MNH, SI | CHE |
| Theridiidae | Phylloneta | impressa | ARA0428 | SPSLO469-13 | KX039343 | NMBE | CHE |
| Theridiidae | Phylloneta | sisyphia | 00786364 | SPSLO274-12 | KX039344 | MNH, SI | CHE |
| Theridiidae | Phylloneta | sisyphia | ARA0416 | SPSLO460-13 | KX039345 | NMBE | CHE |
| Theridiidae | Platnickina | tincta | 00786380 | SPSLO167-12 | KX039353 | MNH, SI | SVN |
| Theridiidae | Robertus | lividus | ARA0201 | SPSLO363-13 | KX039360 | NMBE | CHE |
| Theridiidae | Robertus | mediterraneus | 00786334 | SPSLO275-12 | | MNH, SI | CHE |
| Theridiidae | Robertus | mediterraneus | 00786433 | SPSLO168-12 | | MNH, SI | CHE |
| Theridiidae | Robertus | scoticus | 00786290 | SPSLO320-13 | | MNH, SI | SVN |
| Theridiidae | Robertus | truncorum | 00786435 | SPSLO169-12 | KX039361 | MNH, SI | CHE |
| Theridiidae | Robertus | truncorum | ARA0280 | SPSLO381-13 | KX039362 | NMBE | CHE |
| Theridiidae | Sardinidion | blackwalli | 00786271 | SPSLO321-13 | KX039365 | MNH, SI | SVN |
| Theridiidae | Simitidion | simile | 00786549 | SPSLO170-12 | KX039375 | MNH, SI | SVN |
| Theridiidae | Simitidion | simile | ARA0442 | SPSLO481-13 | KX039374 | EZ LAB | SVN |
| Theridiidae | Steatoda | bipunctata | 00786325 | SPSLO276-12 | KX039380 | MNH, SI | CHE |
| Theridiidae | Steatoda | bipunctata | ARA0029 | SPSLO328-13 | KX039379 | EZ LAB | SVN |
| Theridiidae | Steatoda | triangulosa | 00786489 | SPSLO171-12 | KX039382 | MNH, SI | SVN |
| Theridiidae | Steatoda | triangulosa | ARA0046 | SPSLO334-13 | KX039381 | EZ LAB | SVN |
| Theridiidae | Theridion | betteni | 00786340 | SPSLO277-12 | KX039404 | MNH, SI | CHE |
| Theridiidae | Theridion | pinastri | 00786480 | SPSLO172-12 | KX039406 | MNH, SI | SVN |
| Theridiidae | Theridion | pinastri | ARA0136 | SPSLO351-13 | KX039405 | EZ LAB | SVN |
| Theridiidae | Theridion | varians | 00786374 | SPSLO173-12 | KX039408 | MNH, SI | SVN |
| Theridiidae | Theridion | varians | ARA0043 | SPSLO333-13 | KX039407 | EZ LAB | SVN |
| Thomisidae | Diaea | livens | 00786359 | SPSLO174-12 | KX039138 | MNH, SI | SVN |
| Thomisidae | Ebrechtella | tricuspidata | 00786508 | SPSLO092-12 | KX039157 | MNH, SI | SVN |
| Thomisidae | Ebrechtella | tricuspidata | ARA0033 | SPSLO330-13 | KX039158 | EZ LAB | SVN |
| Thomisidae | Heriaeus | hirtus | 00786469 | SPSLO278-12 | KX039204 | MNH, SI | SVN |
| Thomisidae | Misumena | vatia | 00786387 | SPSLO175-12 | KX039270 | MNH, SI | SVN |
| Thomisidae | Misumena | vatia | ARA0081 | SPSLO342-13 | KX039271 | EZ LAB | SVN |
| Thomisidae | Ozyptila | atomaria | 00786522 | SPSLO176-12 | KX039298 | MNH, SI | CHE |
| Thomisidae | Ozyptila | nigrita | 00786499 | SPSLO093-12 | KX039299 | MNH, SI | SVN |

**Table 1** (*continued*)

| Family | Genus | Species | Sample ID | Process ID | GenBank accession number | Voucher stored at | Collected in |
|--------|-------|---------|-----------|------------|--------------------------|-------------------|--------------|
| Thomisidae | Synema | globosum | 00786485 | SPSLO177-12 | KX039384 | MNH, SI | SVN |
| Thomisidae | Synema | globosum | ARA0390 | SPSLO438-13 | KX039385 | NMBE | CHE |
| Thomisidae | Thomisus | onustus | 00786455 | SPSLO280-12 | KX039410 | MNH, SI | SVN |
| Thomisidae | Thomisus | onustus | ARA0426 | SPSLO467-13 | KX039409 | EZ LAB | SVN |
| Thomisidae | Tmarus | piger | 00786484 | SPSLO178-12 | KX039417 | MNH, SI | SVN |
| Thomisidae | Tmarus | piger | ARA0376 | SPSLO425-13 | KX039418 | EZ LAB | SVN |
| Thomisidae | Xysticus | acerbus | 00786483 | SPSLO179-12 | KX039425 | MNH, SI | SVN |
| Thomisidae | Xysticus | audax | 00786347 | SPSLO180-12 | KX039427 | MNH, SI | SVN |
| Thomisidae | Xysticus | audax | ARA0402 | SPSLO448-13 | KX039426 | EZ LAB | SVN |
| Thomisidae | Xysticus | bifasciatus | 00786543 | SPSLO181-12 | KX039428 | MNH, SI | SVN |
| Thomisidae | Xysticus | cristatus | 00786537 | SPSLO182-12 | KX039430 | MNH, SI | SVN |
| Thomisidae | Xysticus | cristatus | ARA0389 | SPSLO437-13 | KX039429 | NMBE | CHE |
| Thomisidae | Xysticus | desidiosus | 00786372 | SPSLO183-12 | | MNH, SI | SVN |
| Thomisidae | Xysticus | erraticus | 00786275 | SPSLO322-13 | KX039431 | MNH, SI | SVN |
| Thomisidae | Xysticus | kempeleni | 00786486 | SPSLO184-12 | KX039432 | MNH, SI | SVN |
| Thomisidae | Xysticus | kochi | 00786303 | SPSLO323-13 | KX039433 | MNH, SI | SVN |
| Thomisidae | Xysticus | kochi | ARA0434 | SPSLO474-13 | KX039434 | EZ LAB | SVN |
| Thomisidae | Xysticus | lanio | 00786477 | SPSLO185-12 | KX039435 | MNH, SI | SVN |
| Thomisidae | Xysticus | lineatus | 00786535 | SPSLO186-12 | KX039437 | MNH, SI | SVN |
| Thomisidae | Xysticus | lineatus | ARA0304 | SPSLO383-13 | KX039436 | EZ LAB | SVN |
| Thomisidae | Xysticus | macedonicus | 00786376 | SPSLO281-12 | KX039438 | MNH, SI | CHE |
| Thomisidae | Xysticus | tenebrosus | 00786478 | SPSLO187-12 | KX039440 | MNH, SI | SVN |
| Thomisidae | Xysticus | tenebrosus | ARA0332 | SPSLO385-13 | KX039439 | EZ LAB | SVN |
| Titanoecidae | Titanoeca | tristis | 00786297 | SPSLO324-13 | KX039416 | MNH, SI | SVN |
| Uloboridae | Hyptiotes | paradoxus | 00786546 | SPSLO188-12 | KX039216 | MNH, SI | SVN |
| Uloboridae | Hyptiotes | paradoxus | ARA0241 | SPSLO370-13 | KX039215 | EZ LAB | SVN |
| Uloboridae | Uloborus | walckenaerius | 00786324 | SPSLO282-12 | KX039420 | MNH, SI | SVN |
| Zoridae | Zora | spinimana | 00786494 | SPSLO094-12 | KX039449 | MNH, SI | SVN |
| Zoridae | Zora | spinimana | ARA0192 | SPSLO361-13 | KX039448 | NMBE | CHE |

**Notes.**

MNH, SI, National Museum of Natural History, Smithsonian Institution; EZ LAB, Evolutionary Zoology Lab; ZRC SAZU; NMBE, Naturhistorisches Museum der Burgergemeinde Bern; SVN, Slovenia; CHE, Switzerland; MYS, Malaysia.

0.3 μM of each primer, 0.5 mM dNTPs, and 5 units of Biolase DNA polymerase (Bioline). PCR cycling conditions were as follows: 35 cycles of 30 s at 95 °C, 30 s at 48 °C, 45 s at 72 °C. PCR products were cleaned with ExoSAP-IT (Affymetrix), Sanger sequenced using Big Dyes (Life Technologies) and run on a 3730xl DNA sequencer (Applied Biosystems). Sequences were examined for quality and trimmed to the standard barcode segment (649 bp) using Sequencer 5.01 (Gene Codes).

## Barcode library

While we targeted 649 bp long DNA barcodes we also submitted (Table 1) 18 shorter fragments (>570 bp) that still satisfy the requirements of The Barcode of Life Data System

BOLD systems (*Ratnasingham & Hebert*, *2007*). We combined the 297 species barcodes from this study with publically available Araneae sequences from BOLD retrieved 4 December 2013, for a total of 816 species sequences, which formed the test library for this study. Sequences from BOLD were initially included if the sequence length was at least 600 bases and identification was to species. We further filtered and curated the data to exclude sequences whose identification was anonymous or by non-arachnologists, diverged dramatically from all other spider sequences, or for other reasons the sequences were not deemed to be reliable. After having discarded the above, we did not assess the accuracy of every remaining sequence, as it is well known that both BOLD and GenBank contain errors of various kinds, and we wanted our test library to reflect real world conditions. A single sequence was chosen per species from BOLD using these criteria and added to the original sequences from this project, resulting in 816 species representing 313 genera and 49 families (Table 1 and Table S2). Eighteen sequences were singletons at the family level; the maximum number of species per family was 224. 157 sequences were singletons at the genus level; the maximum number of species per genus was 34.

The standalone BLAST+ suite 2.2.28 (*Altschul et al.*, *1990*; *Zhang et al.*, *2000*) was used to create a custom BLAST database from these sequences. Each sequence was then queried against the full set using blastn (MegaBLAST task, minimum e value of 1e–10, maximum of top ten hits other than the hit of the query to itself). For each hit the percent of identical nucleotides in the aligned region (PIdent) was calculated by BLAST. An advantage of using BLAST is the local nature of the alignment hits returned. This will account for differences in sequence lengths in the dataset, which may otherwise affect pairwise identity calculations of complete alignments. A possible outcome of BLAST results are short aligned regions that have high similarity but omit much of the queried sequence. To investigate this, we compared lengths of aligned regions with query sequence lengths to determine the prevalence of this in this dataset. Custom Python scripts (GitHub https://github.com/mkweskin/spider-blast) were used to parse the results, removing the match of the query to itself and to score whether hits matched the genus and family of the query sequence or not. Obviously, if the generic identification matched, the family identification also matched; families therefore always match more often than genera.

On the other hand, singleton generic sequences cannot match correctly at the genus level (for spiders or other poorly known diverse groups), and, likewise, singleton family sequences cannot match correctly at the family level (for spiders or other poorly known diverse groups). We included singletons as targets in order to model more realistically BLAST searches against the BOLD database (many sequences in BOLD are higher level singletons), and also to test more strongly the ability of sequences with two or more species per either genus or family to match correctly. Including 18 singleton family sequences and 157 singleton genus sequences, therefore, increases the probability of misidentification at either ranks and more strongly tests the usefulness of barcodes as supraspecific identification tools.

However, because the 18 unique family sequences must fail at both the family and genus levels, and the 157 unique genus level sequences must fail at the genus level, these necessary

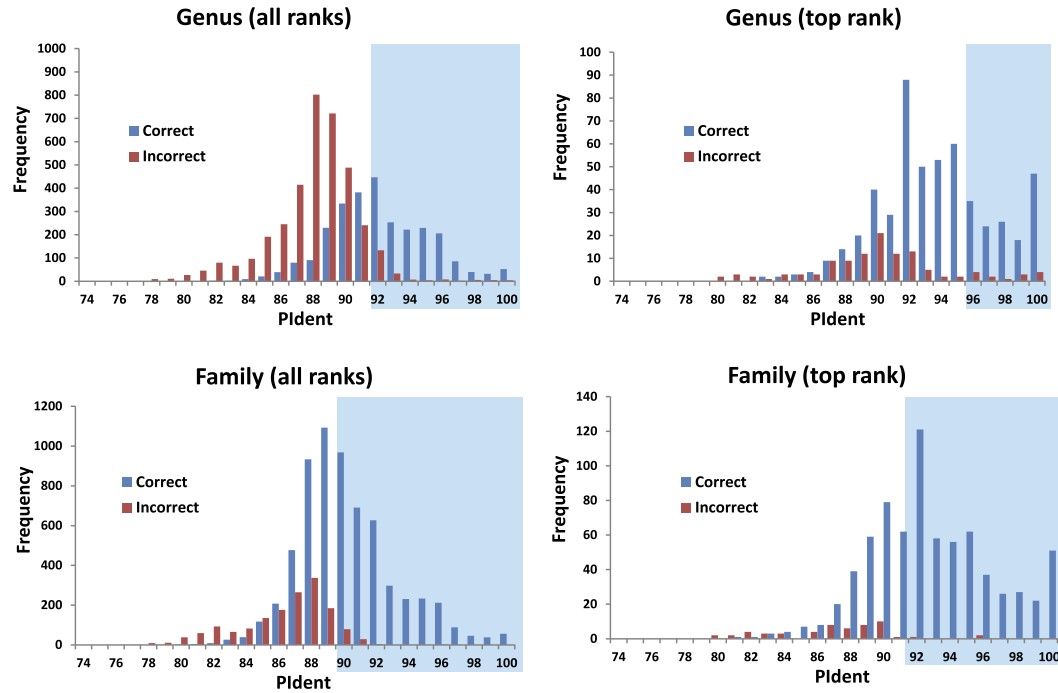

**Figure 2  Results from the barcode matching test.** Frequency distributions of correct and incorrect identifications by percent sequence identity (PIdent) for the top ten and/or best hits at the genus and family level. Shaded areas include hits where no more than 5% of identifications were incorrect.

failures were not included in the overall assessments of the ability of barcode sequences to provide accurate identifications at supraspecific levels.

## RESULTS

The 816 query sequences returned 8,159 total hits with one query only returning nine hits and all others ten (Table S1). PIdent scores ranged from 75% to 100%. We also examined the length of the sequence matched compared to the entire sequence length. 8,114 hits (>99%) matched to 90% or more of the query sequence length indicating that these results represent matches to large portions of the query validating the use of Percent Sequence Identity in the BLAST hits rather than computing the value for a global alignment between sequences. Figure 2 shows the frequency distributions of PIdent values of correct and incorrect identifications at the genus and family rank.

1. 95% of incorrect genus identifications were below PIdent = 95 when all hits for all queries are included, which suggests the latter value as a heuristic threshold to delimit incorrect from correct identifications (for these data). For only the highest rank hits whose PIdent $\geq$ 95, 98% of genus identifications were correct.

2. 95% of incorrect family identifications were below PIdent = 91 when all hits for all queries are included, which suggests the latter value as a heuristic threshold to delimit incorrect from correct identifications (for these data). For only the highest rank hits whose PIdent $\geq$ 91, 97% of family identifications were correct.

3. Library accuracy is crucial, but sequencing, labelling, and identification errors are difficult to detect *a priori*. The highest ranked incorrect family identification was *Meta menardi* (Tetragnathidae) to *Steatoda grossa* (Theridiidae), at PIdent = 96. Further study of the *M. menardi* sequence shows that the BOLD record is probably a mislabeled *Steatoda*. The first true incorrect family identification occurs at a PIdent value of 88; the best hit for *Octonoba* (Uloboridae) is *Amaurobius* (Amaurobiidae).

4. For the 136 genera with at least two species in the library, 76% ($n = 103$) best matched congeners. Thirty-three failed, perhaps because sequences were incorrectly identified taxonomically, or the sequence itself may be erroneous, or perhaps due to non-monophyly of genera.

5. The distributions of PIdents for correct family and genus identifications differ significantly from the distributions of incorrect identifications (Fig. 2).

6. Plotted against increasing numbers of species/genus, and genera/family, the proportion of top ten PIdent values that exceed the above suggested threshold values increases. Roughly speaking, 15 species per genus, and 5 genera per family, are sufficient to ensure that best hits represent correct identifications (Fig. 3).

## DISCUSSION

We show that standard DNA barcodes can accurately assign unknown specimens to genus and family given sufficient sequence identity and sufficient taxonomic representation in the database. Accurate identification (PIdent above which less than 5% of identifications were incorrect) occurred for genera at PIdent values > 95 and families at PIdent values ≥ 91, suggesting these as heuristic thresholds for generic and familial identifications in spiders (shaded in Fig. 2). Accuracy of identification increases with numbers of species/genus and genera/family; above five genera per family and 15 species per genus all identifications were correct (Fig. 3).

The accurate identification of specimens remains a critical challenge for megadiverse groups such as arthropods, most other invertebrates, plants, fungi, protists etc. Morphological identification to species, or even more inclusive taxonomic ranks like genera and families, in many cases requires extensive training, and for most groups taxonomic expertise is limited and dwindling—the so called 'taxonomic impediment' (*Rodman & Cody*, *2003*; *Agnarsson & Kuntner*, *2007*). DNA barcodes have been proposed as convenient tools to overcome this impediment by making identification a purely technical procedure available to any interested researcher or even 'citizen scientists.' However, the accuracy of such a tool strongly depends on the scope and quality of the barcode library (*Smit, Reijnen & Stokvis*, *2013*). Currently available data on databanks like BOLD and GenBank are extensive for some groups, yet the vast majority of species on earth have not yet been barcoded, much less discovered and described taxonomically—each of these tasks is enormous. Even for existing barcoding data, numerous sequences lack accurate taxonomic identification (*Collins & Cruickshank*, *2013*), limiting their utility (e.g., only 58% of Araneae in BOLD are identified to species, and of those many are not correctly identified, as shown in our results; see also *Shen, Chen & Murphy*, *2013*; *Blagoev et al.*, *2016*). Therefore, the

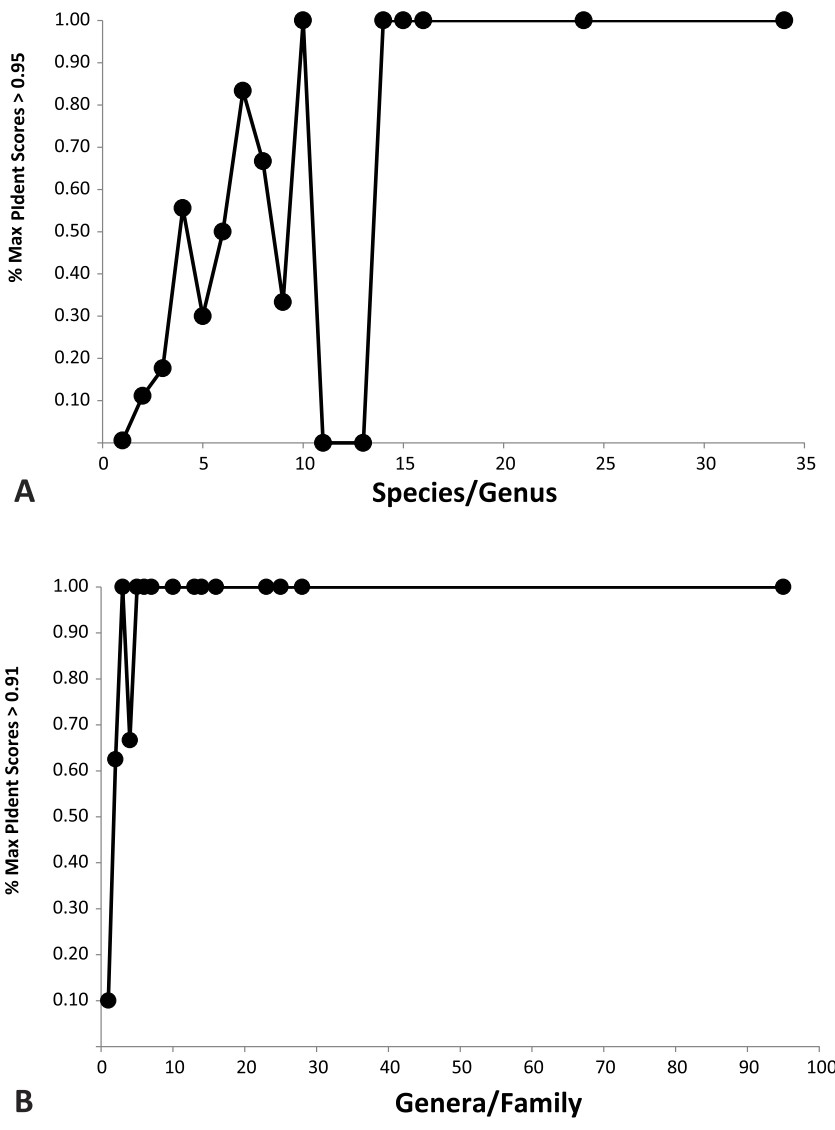

**Figure 3** **Importance of library representation.** Relation between proportion of best sequence identity and numbers of species per genus (A), and genera per family (B). Heuristic thresholds to delimit incorrect from correct identifications were 95 and 91 for genus and family, respectively.

identification of unknown specimens through blasting against BOLD or GenBank will be inaccurate if the databases lack close hits or contain errors. While the ideal database would allow species-level identification by containing barcodes from expertly identified and vouchered specimens of all species, we hypothesized that rapid surveys of well-known biotas can help quickly to build valuable tools allowing identification of larger clades such as genera and families.

Although we were careful to screen available barcode sequences from BOLD to produce a test library with as few errors as possible, it is certainly possible that errors remained, either due to mistakes in the lab or taxonomic identifications of vouchers. For example, *Meta menardi* (Tetragnathidae) blasted to *Steatoda grossa* (Theridiidae) at PIdent = 96, and

BLAST searches on GenBank suggest this *Meta* sequence is actually a *Steatoda*. Likewise, the linyphiids *Agyneta orites* and *Incestophantes frigidus* sequences were identical; one of these records is probably wrong. These sorts of errors bias identifications and limit utility of barcodes. Other examples of identical barcode sequences were all congeners, and therefore are less likely to involve errors but could indicate faults in taxonomy: *Arctosa maculata* and *A. fulvolineata*, *Bolyphantes luteolus* and *B. alticeps*, *Pardosa alacris* and *P. trifrons*, and *Pityohyphantes tacoma* and *P. cristatus*. Likewise, the genus *Neriene* (Linyphiidae) seems non-monophyletic and identifications were thus not accurate.

## CONCLUSIONS

These results suggest that accurate assignment of unknown taxa to genus and family is feasible through DNA barcoding. Database quality is crucial. Numbers of potential matches at generic and familial ranks also affect the probability that an unknown sequence will blast best to the correct family or genus. Unlike the inventory of species, biological discovery of family-level clades of life also seems far advanced—few eukaryotic families, apparently, remain to be discovered. Taken together, these results suggest that barcode-targeted sequencing of exemplars from all families of life (and most genera, if possible) should be an important scientific priority. It would enable approximate taxonomic identification of any organism anywhere on Earth by rapid, cheap, purely technical procedures requiring no specialist knowledge—certainly an important milestone in the on-going attempt to discover, classify, and understand the Earth's biota.

## ACKNOWLEDGEMENTS

Portions of the laboratory and/or computer work were conducted in and with the support of the L.A.B. facilities of the National Museum of Natural History.

### Funding

This work was made possible by a Swiss Contribution to the enlarged EU grant to M Kuntner and C Kropf and the Laboratories of Analytical Biology, National Museum of Natural History, Smithsonian Institution. The funders had no role in study design, data collection and analysis, decision to publish, or preparation of the manuscript.

### Grant Disclosures

The following grant information was disclosed by the authors:
Swiss Contribution to the enlarged EU.
Laboratories of Analytical Biology, National Museum of Natural History, Smithsonian Institution.

### Competing Interests

The authors declare there are no competing interests.

## Author Contributions

- Jonathan A. Coddington and Ingi Agnarsson conceived and designed the experiments, performed the experiments, analyzed the data, wrote the paper, prepared figures and/or tables, reviewed drafts of the paper.
- Ren-Chung Cheng analyzed the data, contributed reagents/materials/analysis tools, reviewed drafts of the paper.
- Klemen Čandek and Matjaž Gregorič contributed reagents/materials/analysis tools, wrote the paper, prepared figures and/or tables, reviewed drafts of the paper.
- Amy Driskell analyzed the data, contributed reagents/materials/analysis tools, wrote the paper, reviewed drafts of the paper.
- Holger Frick, Rok Kostanjšek, Christian Kropf, Tjaša Lokovšek and Nina Vidergar contributed reagents/materials/analysis tools, reviewed drafts of the paper.
- Matthew Kweskin performed the experiments, analyzed the data, wrote the paper, prepared figures and/or tables, reviewed drafts of the paper.
- Miha Pipan contributed reagents/materials/analysis tools, prepared figures and/or tables, reviewed drafts of the paper.
- Matjaž Kuntner conceived and designed the experiments, performed the experiments, analyzed the data, contributed reagents/materials/analysis tools, wrote the paper, prepared figures and/or tables, reviewed drafts of the paper.

## DNA Deposition

The following information was supplied regarding the deposition of DNA sequences:

GenBank accessions are in Table 1 and are also publicly available on BOLD. For all others, see http://ezlab.zrc-sazu.si/dna/.

## Data Availability

Github: https://github.com/mkweskin/spider-blast

## Supplemental Information

Supplemental information for this article can be found online at http://dx.doi.org/10.7717/peerj.2201#supplemental-information.

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
