# Peer review of "DNA barcode data accurately assign higher spider taxa"

_PeerJ, doi:10.7717/peerj.2201_

## Round 0.1 · original submission · Major Revisions

Your manuscript received very different reviews; I suggest that you take more time to justify your approach and contrast it with existing ones. This may still lead to a final rejection (if it becomes clear that your approach does not add anything new to the equation), but on the other hand it may also reveal better the merits of the approach.

A common criticism from both reviewers is the reproducibility; please add much more details.

Although usually not recommended for a revision, we will try to get a third reviewer for the revision who will not have seen the initial submission.

Reviewer 1 ·

Basic reporting

The authors of this MS used BLAST to identify identity thresholds for DNA barcodes to classify them to given families or genera in the database as a proof-of-concept that it is feasible on higher taxonomic level. The english and writing style of the manuscript layout is very good, I do however see conceptional and methodological issues with the paper. Firstly, actually this question has been adressed by various software tools before, and with more sophisticated means, which are not referenced or discussed here. Secondly, the biological definition of families and genera is arbitrary, in contrast to the species concept, so there is no reason why identity levels DNA barcodes should be universally representative for such a level. More details are below.

Experimental design

Major issues:

1) l 105 Definition of what a Family or Genus are, is not properly adressed. These are arbitrarily chosen and there is no biological reason why identity thresholds of any family should be consistent with those of a different one. It is more likely that they are very different between each family, resulting of many ecological, behavioral and evolutionary factors. The idea in general is good to assign higher level taxonomy, but divergence thresholds must be estimated for each taxonomic group separately as they are not transferable even between closely related clades.

2) Other tools already exist that provide higher level taxonomic assignments based on DNA barcodes. Examples are UTAX (Edgar 2010) or RDP (Wang et al. 2007), which can be easily trained for any reference database and have flexible thresholds defined for each group separately. They also provide bootstrap values to have a reliability information for each level. Other tools as e.g. MEGAN (Huson et al. 2007) already provide "least common ancestor" estimations (LCA) from BLAST hits, which is the same approach as described here.

Minor issues:

- l. 131, whole paragraph: this is great that the author have done it, yet a standard in the DNA barcoding community. Many projects do this with multiple thousands of samples without publication, like e.g. the various BoL initiatives worldwide. I do not see this as a central question of an original research article, more of a dataset notification which can be submitted to corresponding journals, e.g. Scientific Data or BMC Research Notes.

- l. 144: "correlated with number of taxa" the result of this question is quite obvious in regard to how BLAST and its databases works. Would be more interesting to correlate "divergence" to see how much methodological error is introduced through the taxonomic structure of the database.

- l. 190: how were they manually edited? Any information to be able to reconstruct this process, e.g. end trimming, phred-scores...

- l. 208 and l. 211: those sentences contradict each other, should be rephrased to clarify what has been done here excactly

- l. 212: which criteria were used to select the representative sequence? Consensus, majority, first?

- l. 236: those can and should be used to adress the false positive rates of the thresholds as done e.g. by Lan et al. 2012

Validity of the findings

Major issues:

3) BLAST is very dependent on the number of reference sequences in the database of the clade of interest, since it has a fixed amount of hits that will be outputted, independently how closely they are related. The authors themselves adress this as one of their results, also. The interpretation of such results generated automatically are thus very dangerous, wherefore the robust classifiers mentioned above are preferable. Also BLAST is a local alignment tool, which may result in only short (perhaps also multiple) sections of the sequences matching references and thus obscuring identity values. The identity should be calculated by semi-global alignments and considering the whole alignment, which is e.g. possible with the EMBOSS package or other tools.

4) Using mitochondrial markers for such a purpose may also not be the ideal choice, the multiple reasons for choosing a nuclear gene on higher level taxonomies and the way of inheritance associated with the genes should be considered and discussed.

Additional comments

This MS does neither bring novel insights into spider taxonomy nor does it provide new tools or concepts for DNA barcoding. I agree that alternative strategies should be on the market, but then they should be compared to existing tools and improvements clearly shown and contrasted, what is not the case here. The results (especially Nr 4.) are rather unconvincing and assumably the classifiers will likely outcompete in most cases when provided with the same database.

I thus cannot recommend publication, the author's should look more at existing literature and tools, and before considering a resubmission evaluate and compare their approach with the other existing ones.

Ancillary:

- l. 161: should also be uploaded to public databases, as e.g. BOLD, EoL or GBIF

- l. 189: Macrogen headquarter is located in Korea

Reviewer 2 ·

Basic reporting

Mostly fine, but there are some minor issues with in-text citations, references and figure legends. Raw data (i.e. sequences) are provided, but the list of additional downloaded records is missing.

Experimental design

The concern about the authors' data curation has been stated in the general comments. Otherwise I am unable to find considerable problems.

Validity of the findings

As mentioned in the manuscript, not much research has been done on this topic so this study brings new and warmly welcomed insights. However, it should be noted that the focus of this study is rather narrow and, thus, the results are not necessary applicable to other groups of animals.

Additional comments

The manuscript of Coddington et al. presents an interesting and needed study on using DNA barcodes to identify spider specimens to higher taxonomic level (i.e. genus or family). The methods are generally sound and the conclusions justified. The manuscript is mainly well-written, though rather compactly at times.

My main concern relates to the set of downloaded sequences from BOLD, which was used as a part of the reference library. To start with, the authors do not provide a list of these specimens so the replication of this study would be impossible based on the current information. I would recommend the authors to add a list including all downloaded specimens in addition to the list of the specimens used in this study. The authors describe their filtering and curating procedure somewhat vaguely (lines 208-210: “We further filtered and curated the data to exclude sequences whose identification was anonymous or by non-arachnologists, diverged dramatically from all other spider sequences, or for other reasons the sequences were not deemed to be reliable”) so it would be important to know which specimens were originally downloaded and subsequently excluded because this potentially has a strong impact on the results. In addition, countries where the specimens of the downloaded records were collected from should be added to the manuscript. It is obvious that the geographical coverage of the dataset affects the results as the amount of knowledge on species varies considerably between different parts of the world.

The authors seem to be rather critical towards BOLD with regard to data quantity and quality (e.g. line 229: “many sequences in BOLD are higher level singletons”, 
lines 289-290: only 58% of Araneae in BOLD are identified to species, and of those many are not correctly identified, as shown in our results“). Although the critique is justified in many cases (but not so well in the manuscript), I find it curious that the authors focus their critique mainly on BOLD, only mentioning GenBank a couple of times together with BOLD. If the authors think that BOLD suffers from quality problems, why did they choose to download their sequences from BOLD instead of GenBank? After all, one might think that GenBank suffers from even higher number of misidentified specimens because the records are not associated with photos and, thus, controlling the quality depends mainly on sequence-based methods.

Minor comments:
- Line 154: The authors say that the specimens were “expertly identified”, but how this was done?
- I would recommend the authors to create a BOLD dataset with DOI for the barcode records used in this study and upload the associated photos to BOLD.
- Lines 258-259: “Thirty-three failed, perhaps because sequences were incorrectly identified taxonomically, or the sequence itself may be erroneous.” Why these were not checked as in the case of Meta menardi?
- Please change the in-text citations and references to follow the guidelines of the journal.
- Please improve the explanation of “all ranks” and “top rank” in the legend of Figure 2. Currently, it is difficult to connect “top ten” with “all ranks” and “best hits” with “top rank”.
- Please replace “BOLD specimen page” and “BOLD sequence page” with “Sample ID” and “Process ID”, respectively, in Table 1.
- Figure 3: “thresholds 95 and 91 respectively” Thresholds of what? Please clarify.

---

## Round 0.2 · Major Revisions

I acknowledge that the manuscript has improved compared to the first version. I also believe that it is a substantial piece of work that deserves to be published. The main problem is that the presentation suggests that the results are general and transferable to other sets of taxa, and this is where we do not follow.

The manuscript is a study on 800+ species of spiders, of which almost 300 are newly characterized (via CO1) in this work.

On the methodological level, there is nothing new (just BLAST).

Title, abstract, introduction and conclusion should accurately reflect the scope of this article. Especially the title "DNA barcode data accurately identify higher taxa" suggests general results that simply are not present.

Reviewer 3 ·

Basic reporting

The manuscript is written in a clear way and error-free. The topic and problem is introduced well and references to similar and previous work are provided. The raw data were submitted to Genbank.

The figures could be more consistent. Each part in Figure 3 should be labeled with an uppercase letter or combined into one figure without dots and with colored lines to better resemble the style of Figure 1.

Experimental design

The authors use BLAST to identify a threshold for the percent identity (PIdent) that separates correctly from incorrectly assigned COI barcode sequences on the genus and family level. They calculate these PIdent values for a test library of 816 spider species. It is doubtful, that their findings can be generalized and the thresholds also hold for other groups. For an exhaustive study it should have been conducted for at least a couple of other groups. Due to different diversities of the groups the PIdent values will likely be specific for each group. The title of the manuscript implies, that their findings can be generalized to all groups.

The authors should provide information on the sequencing (for all labs). Which machine was used, what was the read length etc. In the methods part it sounds like Sanger sequencing, but it should be stated somewhere.

Validity of the findings

In general, the demand to build up a reference database that reflects the diversity of each family is a good idea. The authors made a good start by providing sequences and information on 297 spider species.

The conclusions they draw from their results for spiders is too general. See comment above. The authors should here be more careful in their formulations and change the title and conclusions of their manuscript appropriately.

Additional comments

Please find below the comments to the manuscript "DNA barcode data accurately identify higher taxa" by Jonathan A Coddington, Ingi Agnarsson, Ren-Chung Cheng, Klemen Čandek, Amy Driskell, Holger Frick, Matjaž Gregorič, Rok
Kostanjšek, Christian Kropf, Matthew Kweskin, Tjaša Lokovšek, Miha Pipan, Nina Vidergar and Matjaž Kuntner.

The authors present in the manuscript an approach using percent sequence identity between barcode sequences as a feasible and reasonably accurate method to identify animals to family and genus level. They test their approach using blast searches with CO1 for 816 species of spiders.

Major comments:
In general, there are different understandings and definitions used for the term barcoding and its aim, which is specimen identification and species discovery (see Collins and Cruickshank 2013, "The seven deadly sins of DNA barcoding"). The classical and mostly cited definition is to use a DNA sequence for the identification of a specimen in a sample. The barcode for it has been sequenced before and is available in a reference database. Many researchers will therefore be discontent with the term barcoding for an identification on family level and perhaps 'taxonomic assignment' using CO1 or a similar term would be more appropriate.

Other major comments see Experimental Design and Validity of the Findings.

Minor comments:
Line 228: How are incorrect assignments obtained for a family that has more then 10 species and all 10 BLAST hits for each sequence from this family fall into it? Perhaps considering more then 10 BLAST hits is more appropriate.

Line 230: The local alignment has also disadvantages if the alignments are short but have a high identity for this part. Perhaps a minimal alignment length should be set.

Line 238: For spiders this finding might be correct, but it should be made clear that this might not be the case for all (more diverse) groups.

---

## Round 0.3 · accepted · Accept

As editor, I have no additional comments, and congratulate the authors on the acceptance of their article.

Reviewer 3 ·

Basic reporting

No Comments

Experimental design

No Comments

Validity of the findings

No Comments

Additional comments

The authors have revised their manuscript appropriately and slightly reformulated the most controversial parts of the manuscript. I have no additional remarks or demands for changes.